# SupCLAP: Controlling Optimization Trajectory Drift in Audio-Text Contrastive Learning with Support Vector Regularization

**Jiehui Luo**[2]*, **Yuguo Yin**[1]*[†], **Yuxin Xie**[1], **Jinghan Ru**[1], **Xianwei Zhuang**[1],
**Minghua He**[1], **Aofan Liu**[1], **Zihan Xiong**[4], **Dongchao Yang**[3]
[1]Peking University, [2]Central Conservatory of Music, [3]The Chinese University of Hong Kong,
[4]University of Electronic Science and Technology of China
`luojiehui@mail.ccom.edu.cn, yuguoyin2002@gmail.com`

## Abstract

Contrastive language-audio pretraining, which aims to unify multimodal representations in a shared embedding space, serves as a cornerstone for building a wide range of applications, from cross-modal retrieval to cutting-edge multimodal large language models. However, we find that the perpendicular component of the pushing force from negative samples in contrastive learning is a double-edged sword: it contains rich supplementary information from negative samples, yet its unconstrained nature causes optimization trajectory drift and training instability. To address this, we propose Support Vector Regularization (SVR), a method that introduces an auxiliary support vector to control this perpendicular component, aiming to harness its rich information while mitigating the associated trajectory drift. The efficacy of SVR is critically governed by its semantic radius, for which we explore two unsupervised modeling strategies: direct parameterization and an adaptive radius predictor module enhanced with constraints to improve its predicting accuracy. Extensive experimental results demonstrate that our method surpasses widely used baselines like InfoNCE and SigLIP loss across classification, monolingual retrieval, and multilingual retrieval on standard audio-text datasets. Both the theoretical analysis and the experimental results on optimizing trajectory drift validate the correctness and effectiveness of our SVR method. Notably, our method is highly efficient, it operates without the need for extra training data or inference computation, and adds only a negligible overhead to the training.

## 1 Introduction

Contrastive Language-Audio Pretraining (CLAP) Wu et al. (2023); Ghosh et al. (2025) aims to learn a unified audio-text embedding space by pulling corresponding pairs closer and pushing others apart. This paradigm, which powers applications like cross-modal retrieval Xie et al. (2024) and multimodal LLMs Xue et al. (2024); Lam et al. (2025), has achieved great empirical success. However, standard InfoNCE-based CLAP methods still struggle to learn ideal representations, facing limitations such as poor temporal alignment of audio events Yuan et al. (2024) and inconsistent multilingual alignment Yin et al. (2025). Therefore, achieving optimal alignment between the language and audio representation spaces remains an open challenge.

In this paper, we uncover a complex yet overlooked dynamic in the optimization process of standard InfoNCE-based contrastive learning Wu et al. (2021): optimization trajectory drift. We conceptualize the contrastive learning process as an interplay between a "pulling force" from positive pairs and a "pushing force" from negative pairs within the embedding space. According to the analysis in Section 2 and the experimental results in Figure 1, we find that this pushing force is generally not collinear with the pulling force. This phenomenon stems from both the inherent structure of the data distribution and the stochasticity of mini-batch sampling. The resulting perpendicular component of

---

*Jiehui Luo and Yuguo Yin contributed equally.
[†]Yuguo Yin is the corresponding author.

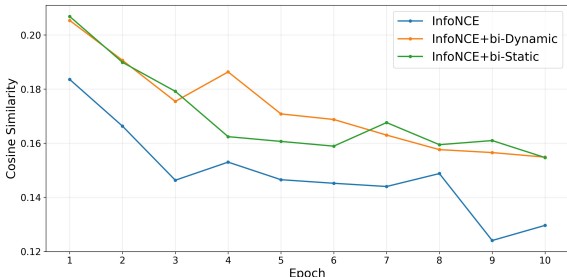

Figure 1: **Optimization Trajectory Drift Analysis.** Drift is measured by the cosine similarity between the update vector and the 'pulling force' vector; a higher similarity indicates lower drift. Compared to InfoNCE loss, our SVR method effectively mitigates this drift. This result confirms the existence of optimization trajectory drift.

the pushing force is a double-edged sword. On the one hand, it contains supplementary information from negative samples with rich learning signals. On the other hand, its uncontrolled and fluctuating nature exerts a sideways force, causing the optimization trajectory to drift. This instability not only slows convergence but also hinders the final alignment quality.

To address the optimization trajectory drift problem, we perform a detailed theoretical and qualitative analysis of the optimization direction of standard InfoNCE-based CLAP methods. Based on this analysis, we design a CLAP framework called SupCLAP, which leverages our proposed Support Vector Regularization (SVR) method. The SVR method introduces a new regularization term to the training objective, formulated as an additional contrastive loss computed between the audio embeddings and newly constructed text support vectors. These support vectors are created by displacing original text embeddings toward their positive audio embedding. The magnitude of this displacement is determined by a semantic radius $R$, which we model in an unsupervised manner. Our analysis demonstrates that SVR effectively reshapes the gradient space, adaptively suppressing the perpendicular component while retaining sufficient supplementary information from negative samples. This yields a more stable optimization trajectory, leading to improved alignment quality.

Building on our analysis for SVR, we further posit that a more precise semantic radius $R$ enables finer control over the perpendicular component, directly enhancing SVR's efficacy. To this end, we explore its unsupervised modeling through two primary strategies. The first strategy is StaticSVR, where the radius is treated as a learnable parameter. The other is the DynamicSVR, which uses an adaptive radius predictor module that utilizes embedding similarity information in the mini-batch to predict the semantic radius $R$. We further propose constraints for the radius predictor module to enhance the quality and stability of the modeled semantic radius. Our main contributions are listed as follows:

- We find and analyze the optimization trajectory drift problem in contrastive learning from a force decomposition perspective, linking it to the perpendicular component of the pushing force from negative samples.

- We propose the SupCLAP scheme with Support Vector Regularization (SVR) and rigorously justify its ability to control optimization trajectory drift. By reshaping the gradient landscape, SVR leads to more stable and effective audio-text contrastive learning.

- We conduct a systematic exploration into unsupervised semantic radius modeling, proposing two strategies: StaticSVR and DynamicSVR. Furthermore, we propose constraints for the DynamicSVR to improve the quality of the predicted radius, validating our hypothesis on the importance of radius precision.

- Extensive experiments on diverse audio-text benchmarks demonstrate that SupCLAP significantly outperforms baseline methods such as InfoNCE and SigLIP loss, notably requiring no additional training data or inference overhead while incurring only negligible training costs. Furthermore, these results provide strong validation for our theoretical analysis of optimization trajectory drift and the effectiveness of SupCLAP.

## 2    ANALYSIS OF OPTIMIZATION TRAJECTORY DRIFT

In this section, we analyze the gradient space of contrastive learning by decomposing its gradient into a pulling force from the positive audio embedding and pushing force from the negative audio embeddings. We then further analyze how the component of the pushing force perpendicular to the pulling force affects model optimization, to clearly illustrate the optimization trajectory drift problem.

### 2.1    CONTRASTIVE LEARNING LOSS FUNCTION AND GRADIENT

First, we define the basic framework for contrastive learning. Assume we have a text embedding $t^+ \in \mathbb{R}^d$, a matching positive audio embedding $a^+ \in \mathbb{R}^d$, where $d$ is the size of embedding, and a batch of $N-1$ mismatched negative audio embeddings $\{a_j^-\}_{j=1}^{N-1}$. To simplify the derivation, we assume all embedding vectors are L2-normalized, i.e., $||t|| = ||a|| = 1$. The similarity function used is the scaled dot product (Scaled Cosine Similarity): $s(a, t) = \cos(a, t)/\tau = \frac{a^T t}{\tau}$, where $\tau$ is the temperature hyperparameter.

The standard InfoNCE loss function Koromilas et al. (2024) (for text-to-audio, as an example) is:

$$L_{orig}(t^+, a^+, \{a_j^-\}) = -\log \frac{\exp(s(t^+, a^+))}{\exp(s(t^+, a^+)) + \sum_{j=1}^{N-1} \exp(s(t^+, a_j^-))}. \tag{1}$$

To analyze the optimization direction, we calculate the gradient of the loss function to the text embedding $t^+$, denoted as $\nabla_t \mathcal{L}_{orig}$. We define:

$$P^+ = \frac{\exp(s(t^+, a^+))}{\exp(s(t^+, a^+)) + \sum_{j=1}^{N-1} \exp(s(t^+, a_j^-))}, \tag{2}$$

$$P_j^- = \frac{\exp(s(t^+, a_j^-))}{\exp(s(t^+, a^+)) + \sum_{j=1}^{N-1} \exp(s(t^+, a_j^-))}, \tag{3}$$

where $P^+ + \sum P_j^- = 1$.

According to the chain rule, $\nabla_t s(t, a) = a/\tau$. The gradient of loss function can be derived as:

$$\begin{aligned}
\nabla_t L_{orig} &= \frac{1}{\tau} \left[ \left( \sum_{j=1}^{N+1} P_j^- a_j^- + P^+ a^+ \right) - a^+ \right] \\
&= \frac{1}{\tau} \left[ (P^+ - 1)a^+ + \sum_{j=1}^{N-1} P_j^- a_j^- \right].
\end{aligned} \tag{4}$$

We decompose the gradient into two parts:

- **Pulling Force**: $F_{pull} = \frac{1}{\tau}(P^+ - 1)a^+$. Since $0 < P^+ < 1$, the term $(P^+ - 1)$ is negative. In gradient descent, we update text embedding $t^+$ in the direction of $-\nabla_t L_{orig}$ (i.e., $t^+ \leftarrow t^+ - \eta \nabla_t L_{orig}$). Therefore, this term is equivalent to a force that pulls text embedding $t^+$ towards positive audio embedding $a^+$.

- **Pushing Force**: $F_{push} = \frac{1}{\tau} \sum_{j=1}^{N-1} P_j^- a_j^-$. This term is a weighted average of all negative example embeddings, and its effect is to push the text embedding $t^+$ away from all negative audio embeddings $a_j^-$. It should also be noted that a negative audio embedding $a_k^-$ with high similarity to the text embedding $t^+$ will yield a larger probability $P_k^-$, thereby exerting a greater pushing force on the text embedding.

### 2.1.1 SUPPRESS TRAJECTORY DRIFT PROBLEM

We find that the pushing force $F_{push}$ is generally not parallel to the direction of the pulling force $F_{push}$. We can decompose the pushing force from a single negative example $a_j^-$, which is given by pushing force $f_{push,j} = \frac{P_j^-}{\tau} a_j^-$. Let us define the unit vector in the pulling direction as $\hat{u} = \frac{a^+ - t^+}{||a^+ - t^+||}$. The negative pushing force $f_{push,j}$ can be decomposed into a component parallel $f_{||,j}$ to $\hat{u}$ and a component perpendicular $f_{\perp,j}$ to $\hat{u}$, which is denoted as $f_{push,j} = f_{||,j} + f_{\perp,j}$. The parallel component is computed as $f_{||,j} = (f_{push,j} \cdot \hat{u})\hat{u}$ while the perpendicular component $f_{\perp,j} = f_{push,j} - f_{||,j} = f_{push,j}(I - \hat{u}\hat{u})$.

During the optimization process, the parallel component $f_{||,j}$ shares the same direction as the pulling force from the positive sample, differing only in magnitude. It thus primarily affects convergence speed and contains little additional information from the negative samples beyond what is already present in the pulling force. In contrast, the gradient direction of the perpendicular component $f_{\perp,j}$ is rich with supplementary information from the negative samples, which is distinct from that of the positive sample. But this perpendicular component acts as a double-edged sword: while its direction provides additional information, its uncontrolled magnitude can cause the optimization path to drift continuously. The nature of this drift can be understood on both global and local levels, which is illustrated in Figure 2. We put the detailed analysis the trajectory drift problem in Appendix C.

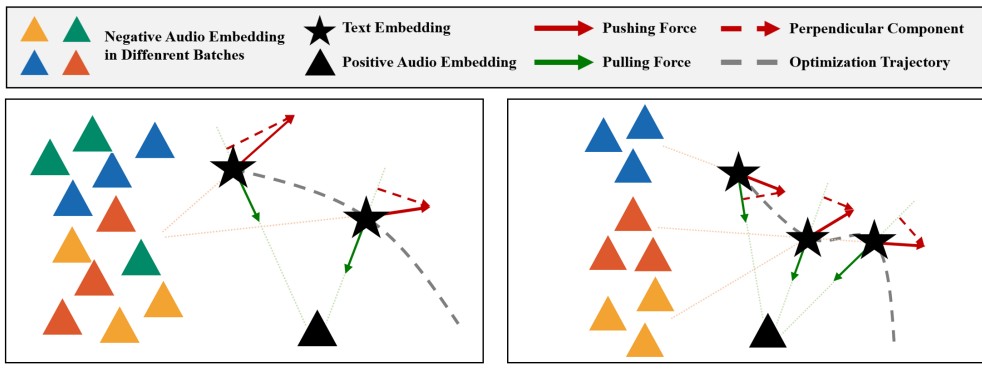

**Figure 2: Illustration of Global and Local Perpendicular Components**. The left subfigure depicts the global perpendicular component. The subfigure on the right illustrates the local perpendicular component. For clarity in demonstrating the local perpendicular component, the negative audio embeddings are shown in the right subfigure with distinct distributions across batches. In practice, the negative distributions across batches are more likely to overlap, as shown in the left subfigure.

- **Global Perpendicular Component**: From a global perspective, even if we could access all negative examples in the dataset, the direction of their weighted resultant force would seldom be collinear with the direction of the pulling force from a specific positive example. This creates a systematic, global perpendicular component. Throughout the entire model training process, this global component continuously pushes the optimization path of the text embedding away from the ideal straight-line trajectory, resulting in a systematic drift in the entire alignment path.

- **Local Perpendicular Component**: In practice, training is performed using a mini-batch strategy, which means that at each update step, the negative examples seen by the model are only a random subset of the total population of negative examples. The distribution of negative examples varies between batches, causing the direction and magnitude of the perpendicular component to change drastically with each update. This uncertain and random sideways push is the direct cause of the local, high-frequency oscillations observed in the optimization path.

Acting in concert, these two perpendicular components give rise to the problem of optimization trajectory drift in audio-text contrastive learning. This reduces the model's convergence efficiency and

ultimately limits the alignment accuracy of the learned representations if the perpendicular components are not well controlled.

## 3 METHOD

To address the optimization trajectory drift identified in our analysis, we propose the SupCLAP framework built around the Support Vector Regularization (SVR) method. The main idea of SVR is to control the perpendicular component of pushing force by using an auxiliary regularization term to align text support vectors and audio embeddings, thereby guiding the optimization towards a more stable and direct trajectory. The construction of this support vector is critically dependent on a semantic radius $R$, which dictates the magnitude of the regularization. As the semantic radius $R$ lacks direct supervision, we explore two unsupervised strategies for modeling the semantic radius: a straightforward direct parameterization (StaticSVR) and a more sophisticated adaptive radius predictor module (DynamicSVR), which is further enhanced with a constraint term to improve the quality of the predicted semantic radius. We analyze SVR's effect mathematically in Appendix D.

### 3.1 FRAMEWORK OF SUPCLAP

The SupCLAP framework is built upon the standard symmetric contrastive learning objective common in CLAP architectures. We begin by defining this baseline. For a given batch, we have a positive audio-text pair $(a^+, t^+)$ and $N$ negative pairs $\{(a_j^-, t_j^-)\}_{j=1}^N$, all represented in a $d$-dimensional embedding space. The affinity between audio and text embeddings is measured by the scaled cosine similarity $s(a, t) = \cos(a, t)/\tau$, where $\tau$ is a temperature hyperparameter. The standard training objective $L_{orig}$ consists of two InfoNCE loss terms: one for text-to-audio alignment loss $L_{orig,t2a}$ and one for audio-to-text alignment $L_{orig,a2t}$. The total original loss is formulated as follows:

$$
\begin{aligned}
L_{orig} =& L_{orig,t2a} + L_{orig,a2t} \\
=& -\log \frac{\exp(s(t^+, a^+)/\tau)}{\exp(s(t^+, a^+)/\tau) + \sum_{j=1}^N \exp(s(t^+, a_j^-)/\tau)} \\
& -\log \frac{\exp(s(a^+, t^+)/\tau)}{\exp(s(a^+, t^+)/\tau) + \sum_{j=1}^N \exp(s(a^+, t_j^-)/\tau)}.
\end{aligned}
\tag{5}
$$

To address the trajectory drift problem, we design the Support Vector Regularization (SVR) term, denoted as $L_{svr}$. The final training objective of SupCLAP is a weighted sum of the original loss and our SVR regularization term:

$$
\begin{aligned}
L_{SupCLAP} =& L_{orig} + \alpha L_{svr} \\
=& L_{orig} - \alpha \log \frac{\exp(s(t_{sup}, a^+))}{\sum_{j=1}^N \exp(s(t_{sup}, a_j^-))},
\end{aligned}
\tag{6}
$$

The text support vector $t_{sup} = t^+ + R\hat{u}$ is constructed by displacing the original text embedding $t^+$ by the semantic radius $R \in \mathbb{R}$ along the unit vector $\hat{u} = \frac{a^+ - t^+}{||a^+ - t^+||}$. The hyperparameter $\alpha$ controls the influence of this SVR term in the final loss function. We will detail the unsupervised strategies for modeling the semantic radius $R$ in the following subsection. The intuition behind this design is that by creating an auxiliary SVR term, it reshapes the gradient space. It is engineered to selectively control the perpendicular component of the pushing force from negative samples, which is the primary source of trajectory drift. A detailed mathematical analysis of SVR's mechanism is provided in Appendix D. We conclude that after adding the SVR term, the parallel component of pushing force from the $j$-th negative audio embedding $a_j^-$ is

$$
F_{||,push,j} = \left( \frac{P_j^-}{\tau} + \alpha \frac{P_{sup,j}^-}{\tau} \right) a_{||,j}^-,
\tag{7}
$$

while the perpendicular component of the pushing force is

$$F_{\perp,push,j} = \left[\frac{P_j^-}{\tau} + \alpha\frac{P_{sup,j}^-}{\tau}\left(1 - \frac{R}{||a^+ - t^+||}\right)\right]a_{\perp,j}^-. \tag{8}$$

The probability $P_j^-$ and $P_{sup,j}^-$ can be calculated following equation 3 using text embedding $t^+$ and text support vector $t_{sup}$ separately. $a_{||,j}^-$ denotes the parallel component of audio embedding $a_j^-$ to the unit vector $\hat{u} = \frac{a^+ - t^+}{||a^+ - t^+||}$, while $a_{\perp,j}^-$ denotes the perpendicular component of audio embedding $a_j^-$ to the unit vector $\hat{u}$. The SVR term works by reshaping the total pushing force: while the parallel component is preserved, the perpendicular component is uniquely scaled by the factor $(1 - \frac{R}{||a^+ - t^+||})$. This factor allows SVR to selectively suppress the magnitude of the perpendicular component of the pushing force. A larger semantic radius R leads to a higher degree of suppression on this component. By exerting this precise control, SVR effectively controls the trajectory drift while harnessing the rich information from negative samples.

**Note:** Although our illustration and analysis of SVR are focused on the unidirectional (text-to-audio alignment) SVR for clarity, the principle is not exclusive to this direction. As demonstrated by the experimental results in Subsection 4.2, incorporating bidirectional SVR in both text-to-audio and audio-to-text alignments yields superior performance.

**Inference pipeline:** SupCLAP's inference pipeline is identical to that of standard CLAP methods Elizalde et al. (2023). Retrieval and classification are performed solely by ranking the embedding similarity between audio and text, without the need to compute auxiliary support vectors like $t_{sup}$.

## 3.2 UNSUPERVISED SEMANTIC RADIUS MODELING

The effectiveness of our proposed SVR critically depends on the semantic radius $R \in \mathbb{R}$. This parameter is central to our method as it directly governs the factor $(1 - \frac{R}{||a^+ - t^+||})$, which scales the perpendicular component of the pushing force. Since datasets provide no ground-truth values for the semantic radius $R$, we frame its estimation as an unsupervised modeling problem. To this end, we propose and investigate two distinct strategies:

- **Static modeling (StaticSVR)**: The StaticSVR is a modeling strategy focused on suppressing the global perpendicular component. We model the semantic radius R as a single, globally shared, learnable scalar. This scalar is jointly optimized with other model parameters to minimize the total loss $L_{SupCLAP}$. For any text embedding $t^+$, this constant radius is used to construct the support vector $t_{sup} = t + R \cdot \hat{u}$, where $\hat{u}$ is the unit vector of the pulling force.

  This method's primary advantages are its simplicity and the stability of the global radius $R$. However, its limitation is the idealized assumption that a single, constant radius is optimal for all instances. This static approach lacks the flexibility to handle the varying complexity and alignment difficulty among different audio-text pairs.

- **Dynamic modeling (DynamicSVR)**: The DynamicSVR is a modeling strategy focused on suppressing the local perpendicular component. For adaptive, instance-specific control, we propose a radius predictor. We hypothesize the optimal semantic radius $R$ is not fixed; it should adapt dynamically to the local embedding geometry, defined by the relationship between a text embedding $t^+$ and its positive and negative audio counterparts. To learn this relationship, we design a radius predictor—a lightweight Multi-Layer Perceptron (MLP) $f_\theta$ which learns to output a suitable radius in a fully unsupervised manner. Formally, the predictor $f_\theta : \mathbb{R}^N \rightarrow \mathbb{R}$ maps a local similarity vector $S$ to an instance-specific semantic radius $R = f_\theta(S)$. The similarity vector $S$ is composed of $S = [s(t^+, a^+), s(t^+, a_1^-), ..., s(t^+, a_{N-1}^-)]$.

  The similarity vector $S$ informs the radius predictor by capturing local embedding geometry. Each values in $S$ measures the proximity between text embedding $t^+$ and audio embeddings, collectively signaling potential trajectory drift; for instance, high similarity to a negative sample implies a higher drift risk. Learning these patterns allows the predictor to estimate a custom semantic radius $R$ for precise control. This adaptive approach's main advantage is its flexibility to adjust the optimization trajectory to each mini-batch's alignment

difficulty. However, DynamicSVR's performance is heavily dependent on the prediction accuracy of the batch-level radius $R$. Inaccurate predictions of $R$—caused by noisy data, a noisy embedding space, or a weak pretrained model—may make DynamicSVR's performance inferior to that of the simpler StaticSVR.

We propose a constraint term $L_{cons}$, for the radius predictor module to mitigate the problem of the predicted semantic radius R being unstable or inaccurate. Due to space limitations, the introduction of this constraint term is provided in Appendix B.

# 4 EXPERIMENT

In this section, we present a comprehensive evaluation of our proposed SupCLAP framework. We begin by detailing the datasets, models, evaluation metrics, and experimental setup. We then benchmark our main results on monolingual and multilingual audio-text retrieval, as well as on the zero-shot classification tasks. Subsequently, we conduct detailed ablation studies to dissect the contribution of each component in our method, the effectiveness of SVR under different batch sizes, and the impact of the SVR term's weight $\alpha$. Finally, we evaluate the additional time and GPU memory overhead and show that they are negligible. In addition, we experimentally analyze how the modeled semantic radius changes with training epochs. These experiments further demonstrate the theory of trajectory drift and the effectiveness of our proposed approach. Due to space limitations, the evaluation results and corresponding analysis on the multilingual dataset, partial ablation study results, and the analysis of SVR's overhead are presented in Appendix E.

## 4.1 EXPERIMENTAL SETUP

### 4.1.1 DATASETS AND METRICS

Our experiments are conducted on the AudioCaps Kim et al. (2019) and Clotho Drossos et al. (2020) datasets. AudioCaps consists of approximately 49k 10-second audio clips, where each training sample is paired with a single caption, and each validation/test sample has five captions. Clotho contains 6974 clips of 15-30 seconds in duration, all of which are annotated with five captions. For our multilingual scenario, we augment these datasets by translating all English captions into seven other languages (fra, deu, spa, nld, cat, jpn, zho). All audio clips are resampled to 16 kHz.

We evaluate retrieval performance in both monolingual and multilingual settings using Recall at rank k (**R@k**), which is 1 if the target is found in the top-k results, and mean Average Precision at 10 (**mAP10**), which evaluates precision scores across all queries for the top 10 retrieved items.

### 4.1.2 MODELS AND IMPLEMENTATION DETAILS

**Model Architecture:** Our framework for audio-text retrieval leverages two powerful pretrained encoders to handle the respective modalities. We adopt the CED-Base model Dinkel et al. (2024) as our audio encoder. We employ the multilingual SONAR-TE model Duquenne et al. (2023) as text encoder, which has demonstrated good performance on cross-lingual similarity benchmarks like xsim and xsim++, making it well-suited as the text encoder for this paper. The sentence embeddings are computed by pooling the encoder's token-level hidden states. The semantic radius predictor is implemented as an MLP with 3 layers.

**Implementation Details:** To ensure a fair and controlled comparison, all models are initialized with weights from CED-Base and SONAR-TE and trained for 10 epochs on a single NVIDIA H800 GPU. The training employs the Adam optimizer with a learning rate of $5 \times 10^{-5}$. The batch size is set to 24, and a temperature of $\tau$=0.07 for the contrastive loss. The SVR weight $\alpha$ is set to 1. We investigate two distinct audio-text retrieval scenarios: a monolingual scenario, utilizing only English captions, and a multilingual scenario. The model checkpoint yielding the highest recall on the test set is selected for final evaluation. We name our SVR methods like "bi-DynamicSVR" and "uni-StaticSVR". The prefix "bi" or "uni" indicates bidirectional (both audio-to-text and text-to-audio) or unidirectional (text-to-audio) SVR, respectively. The suffix "StaticSVR" or "DynamicSVR" denotes the different ways of modeling the semantic radius $R$ mentioned in 3.2.

## 4.2 Main Results

### 4.2.1 Audio-Text Retrieval

To evaluate the effectiveness of our proposed framework, we compare four methods: (1) the standard **InfoNCE** loss; (2) the standard **SigLIP** loss Zhai et al. (2023); (3) our **bi-StaticSVR**, where the semantic radius $R$ is a single, globally shared, learnable parameter; and (4) our **bi-DynamicSVR**, which utilizes an adaptive predictor to estimate an instance-specific radius $R$.

Table 1: Recall and precision results under monolingual AudioCaps and Clotho dataset

| Model | AudioCaps | | | | Clotho | | | |
|---|---|---|---|---|---|---|---|---|
| | T2A | | A2T | | T2A | | A2T | |
| | R@1 | R@10 | R@1 | R@10 | R@1 | R@10 | R@1 | R@10 |
| MMT | 36.10 | 84.50 | 39.60 | 86.70 | 6.70 | 33.20 | 7.00 | 34.60 |
| ML-ACT | 33.90 | 82.60 | 39.40 | 83.90 | 14.40 | 49.90 | 16.20 | 50.20 |
| CLAP | 34.60 | 82.00 | 41.90 | 84.60 | 16.70 | 54.10 | 20.00 | 58.70 |
| CompA-CLAP | 36.10 | 81.60 | 45.20 | 86.70 | 16.80 | 56.10 | 19.70 | 55.60 |
| LAION-CLAP | 34.50 | 80.20 | 42.50 | 87.40 | 15.80 | 52.90 | 19.10 | 54.90 |
| GPA | 36.20 | 82.90 | 44.20 | 86.70 | 15.70 | 50.90 | 18.60 | 55.30 |
| T-CLAP | 39.70 | 86.90 | 49.80 | 91.90 | 17.30 | 53.60 | 21.80 | 57.40 |
| ReCLAP | 37.10 | 85.00 | 48.00 | 90.80 | 18.90 | 59.00 | 20.50 | 58.90 |
| Cacophony | 41.00 | 86.40 | 55.30 | 92.40 | 20.20 | 58.80 | 26.50 | 67.30 |
| SigLIP | 36.74 | 85.71 | 48.00 | 88.03 | 13.58 | 51.21 | 17.10 | 52.56 |
| -bi-StaticSVR | 42.54 | 87.61 | 55.25 | 90.55 | 16.21 | 53.60 | 21.26 | 59.13 |
| -bi-DynamicSVR | **43.09** | **89.26** | **56.30** | **92.67** | **17.51** | **56.85** | **22.71** | **60.87** |
| InfoNCE | 41.87 | 87.69 | 56.72 | 92.33 | 18.67 | 58.42 | 22.61 | 63.09 |
| -bi-StaticSVR | 43.89 | 88.78 | 57.77 | 92.75 | 19.50 | 58.86 | 24.93 | 63.19 |
| -bi-DynamicSVR | **44.16** | **89.24** | **59.66** | **93.49** | **19.75** | **59.13** | **25.31** | **63.29** |

**Monolingual Retrieval:** Results from the monolingual audio-text retrieval task, as shown in Table 1, demonstrate that both bi-StaticSVR and bi-DynamicSVR effectively enhance the performance of InfoNCE and SigLIP. This validates our analysis of the perpendicular component in contrastive learning and the effectiveness of our proposed SVR term in resolving the trajectory drift problem. The superior performance of InfoNCE compared to SigLIP stems from its Softmax-based competitive mechanism, which provides a stronger gradient signal for effective discrimination on diverse audio datasets like AudioCaps and Clotho that contain numerous hard negatives.

### 4.2.2 Zero-Shot Audio Classification

We also assess zero-shot audio classification on the ESC-50 and US8K benchmarks using our monolingual model loaded with pretrained weights provided in ML-CLAP Yan et al. (2024). Text labels are constructed using the template "This is a sound of {classlabel}", and we report top-1 accuracy based on the highest cosine similarity between audio and text embeddings.

Table 2: Zero-shot Audio Classification Performance of the CLAP Model

| Model | Audio Classification Dataset & Setting | | |
|---|---|---|---|
| | ESC-50 | US8K | VGGSound |
| Wav2CLIP | 41.4 | 40.4 | 10.0 |
| AudioCLIP | 69.4 | 65.3 | - |
| CLAP | 82.6 | 73.2 | - |
| LAION-CLAP | 89.5 | 76.3 | 23.1 |
| Collat | 84.0 | 77.0 | - |
| InfoNCE | 89.6 | 81.63 | 24.57 |
| -bi-StaticSVR | 90.7 | 83.63 | 24.65 |
| -bi-DynamicSVR | **92.1** | **83.74** | **25.11** |

As shown in Table 2, the bidirectional dynamic SVR achieves the highest classification accuracy. This further demonstrates the generalization capability of SVR, which can learn more robust and semantically meaningful feature representations by effectively suppressing trajectory drift, thereby enhancing the model's performance in both retrieval and classification tasks.

### 4.2.3 ABLATION STUDIES

We conducted extensive ablation studies on the AudioCaps dataset. These studies include evaluating the impact of SVR components with different directions and modeling strategies on model performance, the generalization of SVR under different batch sizes, and the effect of the SVR weight $\alpha$ on model performance. Due to space limitations, the ablation experiment results and analysis for different batch sizes and the SVR weight are presented in Appendix E.10. The results show that both StaticSVR and DynamicSVR can effectively improve the performance of contrastive learning across various batch sizes, and the best results are achieved when the SVR weight $\alpha$ is set to 1.

Table 3: Ablation Study of SVR Variants on Monolingual Text-Audio Retrieval.

| ID | Model | T2A | | A2T | |
|---|---|---|---|---|---|
| | | R@1 | mAP10 | R@1 | mAP10 |
| 0 | InfoNCE | 41.87 | 56.74 | 56.72 | 35.36 |
| 1 | -bi-DynamicSVR | **44.16** | **58.79** | **59.66** | **36.69** |
| 2 | -bi-DynamicSVR wo/ constraints | 44.01 | 58.47 | 59.24 | 36.64 |
| 3 | -uni-DynamicSVR | 43.63 | 58.16 | 58.51 | 36.00 |
| 4 | -uni-DynamicSVR wo/ constraints | 43.53 | 58.11 | 57.67 | 35.96 |
| 5 | -bi-StaticSVR | 43.89 | 58.36 | 57.77 | 35.72 |
| 6 | -uni-StaticSVR | 43.28 | 57.95 | 57.56 | 34.62 |

**Effectiveness of SVR Components:** The results in Table 3 systematically demonstrate the effectiveness of our proposed components for SVR. The fully-equipped bidirectional DynamicSVR model with constraints (bi-DynamicSVR) achieves the best results. We find that the unidirectional SVR outperforms the baseline model. This performance gain is further amplified when bidirectional SVR is simultaneously applied to both audio-to-text and text-to-audio directions. Furthermore, the experimental results indicate that introducing the constraint term in equation 9 improves the accuracy of DynamicSVR in modeling the semantic radius $R$, further enhancing the effectiveness of SVR.

### 4.3 SEMANTIC RADIUS ANALYSIS

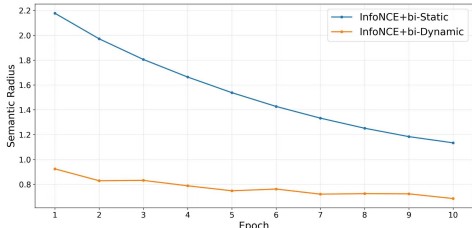

Figure 3: Results of Semantic Radius Changes

We extract the parameterized radius for bi-StaticSVR at each epoch and compute the average predicted radius for bi-DynamicSVR. As depicted in Figure 3, the semantic radius $R$ decreases as training progresses. Based on our analysis in Appendix E.8, this phenomenon provides evidence that self-supervised modeling effectively learns the trade-off between suppressing the perpendicular component and preserving information from negative samples. Furthermore, bi-StaticSVR's radius curve is smoother, reflecting its stable, global modeling of the perpendicular component, whereas bi-DynamicSVR's local approach results in greater fluctuations across batches.

## 5 CONCLUSION

This work addresses optimization trajectory drift in audio-text contrastive learning, an instability caused by the perpendicular component of the pushing force from negative samples. We propose SupCLAP, a framework that incorporates Support Vector Regularization (SVR) to mitigate this drift. SVR introduces an auxiliary support vector to reshape the gradient space, controlling the perpendicular force and stabilizing the optimization path. We explore two unsupervised strategies for its key

parameter, the semantic radius: StaticSVR and DynamicSVR. Experiments show that SupCLAP significantly outperforms InfoNCE baselines on retrieval and classification tasks with negligible computational overhead, validating our approach's effectiveness and practical viability.

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

## A    RELATED WORK

Learning joint representations for audio and text has become a pivotal area of research, enabling applications from cross-modal retrieval Elizalde et al. (2023) to text-to-audio generation Yuan et al. (2025). A dominant paradigm in this domain is contrastive learning, which aims to create a unified embedding space where audio and its corresponding text description are brought closer together, while dissimilar pairs are pushed apart. Drawing inspiration from the success of CLIP Radford et al. (2021); Zhuang et al. (2025b) in the vision-language domain Yu et al. (2022); Zhuang et al. (2025a), the Contrastive Language-Audio Pretraining (CLAP) framework based on InfoNCE loss has emerged as a foundational approach Wu et al. (2022); Guzhov et al. (2022). Researchers have since focused on refining and extending the capabilities of CLAP. For instance, to better manage audio inputs of varying durations and boost overall performance, Wu et al. (2023) integrated a feature fusion mechanism with a keyword-based description enhancement strategy. Others have focused on improving the model's fine-grained understanding. Silva et al. (2023) introduced COLLAT, a framework that achieves nuanced audio comprehension by freezing the language model's parameters and training the audio encoder with a specialized audio-text alignment objective. Similarly, to address the challenge of distinguishing between closely related but distinct sounds, such as hard negative samples. Ghosh et al. (2023) developed a modular contrastive loss designed to improve the model's discriminative capabilities. In subsequent work, Ghosh et al. (2025) further explored the impact of textual data quality, demonstrating that rewriting audio captions to be more descriptive significantly enhances the model's comprehension of real-world acoustic scenes. Beyond these, domain specialization has gained attention. Liu et al. (2024) proposed DSCLAP for domain-specific audio-text pre-training, emphasizing tailored representations for specialized contexts. Meanwhile, Zhu et al. (2024) introduced Cacophony, which strengthens retrieval with auxiliary objectives.

Existing works have explored integrating principles from Support Vector Machine (SVM) Hearst et al. (1998) into contrastive learning from different perspectives. Existing approaches include Max-Margin Contrastive Learning Shah et al. (2022), which adapts the SVM max-margin principle to identify and push away hard negative samples treated as support vectors. SV-Learner Liang et al. (2024) is another approach that employs SVM to select reliable data pairs for contrastive learning, thereby enhancing robustness in noisy-label scenarios. Concurrently, the T-MASS Wang et al. (2024) approach also utilizes a support vector to model text as a stochastic mass, aiming to enrich text's semantic representation for text-video retrieval. Compared to the above approaches, our work SupCLAP introduces Support Vector Regularization (SVR) to address a different core problem: controlling the optimization trajectory drift caused by the perpendicular pushing force from negative samples in contrastive learning.

## B    CONSTRAINT TERM IN DYNAMICSVR

To improve the stability and precision of the predicted semantic radius $R$, we propose a constraint term $L_{cons}$ on the radius predictor module. While the adaptive predictor offers flexibility, an unconstrained radius vector $R$ could lead to two potential failure modes:

- **Excessive Magnitude**: A semantic radius $R$ with an excessively large magnitude (i.e., $R \gg ||a^+ - t^+||$) causes the factor $(1 - \frac{R}{||a^+ - t^+||})$ to become negative. This may lead to a directional inversion of the perpendicular component representing the additional information of the negative samples. As a result, the effective utilization of this information is impeded, which in turn leads to optimization instability.

- **Counterproductive Direction**: Without constraints on the semantic radius $R$ during unsupervised modeling, the radius predictor may output $R<0$, which results in the factor $(1 - \frac{R}{||a^+ - t^+||})>1$. This adversely magnifies the perpendicular component, thereby aggravating the problem of optimization trajectory drift.

To simultaneously address both issues, we formulate the constraint term $L_{cons} = Relu(R - ||a^+ - t^+||) + Relu(-R)$. This term is composed of two components, each targeting one of the aforementioned failure modes. The first $Relu(R - ||a^+ - t^+||)$ directly penalizes large magnitudes to prevent overshooting. The second, $Relu(-R)$, explicitly encourages directional alignment with the pulling

force. By incorporating this constraint, the total loss function becomes:

$$L_{SupCLAP-Cons} = L_{orig} + \alpha L_{svr} + \beta L_{cons}. \tag{9}$$

These constraints ensure that the support vector provides a stable and meaningful regularization signal, transforming SVR into a more robust and effective fine-tuning mechanism. We use a default constraint weight of $\beta = 0.01$, which is intended to impose a slight penalty, ensuring the semantic radius $R$ remains within a reasonable range without dominating the loss function.

## C   DETAILED THEORETICAL AND EMPIRICAL ANALYSIS OF OPTIMIZATION TRAJECTORY DRIFT

In this section, we elucidate the causal relationship between controlling the perpendicular component and performance improvement by examining the two fundamental optimization objectives of contrastive learning: pulling positive pairs closer and pushing negative pairs apart.

### C.1   THEORETICAL PROOF ON CONVERGENCE SPEED OF POSITIVE PAIRS

To demonstrate that the perpendicular component ($F_\perp$) reduces the optimization speed of representation alignment, we analyze the change in the squared Euclidean distance between the anchor text embedding $t^+$ and the positive audio embedding $a^+$. Let $t_{next}^+$ denote the text embedding in the next update step. The update rule is given by $t_{next}^+ = t^+ - \eta F$, where $\eta$ is the learning rate, and the total gradient force is decomposed into parallel and perpendicular components relative to the pulling direction: $F = F_{||} + F_\perp$. We quantify the convergence speed by the reduction in the squared distance to the target $a^+$. A smaller value of the difference $||t_{next}^+ - a^+||^2 - ||t^+ - a^+||^2$ implies a larger reduction in distance, and thus, a faster convergence speed. The derivation is as follows:

$$
\begin{aligned}
||t_{next}^+ - a^+||^2 &= ||(t^+ - a^+) - \eta(F_{||} + F_\perp)||^2 \\
&= ||t^+ - a^+||^2 - 2\eta(t^+ - a^+)^T(F_{||} + F_\perp) + \eta^2||F_{||} + F_\perp||^2.
\end{aligned}
\tag{10}
$$

Since $F_\perp$ is orthogonal to both the pulling direction $(t^+ - a^+)$ and the parallel component $F_{||}$, we have $(t^+ - a^+)^T F_\perp = 0$ and $||F_{||} + F_\perp||^2 = ||F_{||}||^2 + ||F_\perp||^2$. Substituting these into the equation:

$$
\begin{aligned}
||t_{next}^+ - a^+||^2 &= ||t^+ - a^+||^2 - 2\eta(t^+ - a^+)^T F_{||} + \eta^2(||F_{||}||^2 + ||F_\perp||^2) \\
&= ||t^+ - a^+||^2 - 2\eta(t^+ - a^+)^T F_{||} + \eta^2||F_{||}||^2 + \eta^2||F_\perp||^2 \\
&= ||t^+ - a^+||^2 - \eta(2t^+ - 2a^+ - \eta F_{||})^T F_{||} + \eta^2||F_\perp||^2
\end{aligned}
\tag{11}
$$

The change in distance can thus be decomposed into two distinct terms:

$$
\underbrace{||t_{next}^+ - a^+||^2 - ||t^+ - a^+||^2}_{\substack{\text{Optimization dynamics} \\ \text{(convergence speed of positive pairs)}}} = \underbrace{-\eta(2t^+ - 2a^+ - \eta F_{||})^T F_{||}}_{\text{Optimization dynamics along the ideal direction}} + \underbrace{\eta^2||F_\perp||^2}_{\text{Noise penalty}}
\tag{12}
$$

- Optimization along the ideal direction: In the first term, because the learning rate $\eta$ is typically small, the hindrance caused by the second-order parallel component ($\eta^2||F_{||}||^2$) is far smaller than the promotion of convergence speed provided by the first-order parallel component ($2\eta(t^+ - a^+)^T F_{||}$). This term represents the effective progress along the geodesic path.

- Noise penalty: The second term $\eta^2||F_\perp||^2$ depends solely on the perpendicular component. Since this term is strictly greater than 0 if the perpendicular component is not zero, it acts as a penalty that increases the final distance. Therefore, a larger perpendicular component imposes a greater hindrance to the convergence speed of positive pairs. This mathematical conclusion validates our hypothesis that suppressing the perpendicular component accelerates alignment.

C.2 Analysis of the Pushing Objective and Stochastic Deviation

Regarding the objective of pushing negative pairs apart, we analyze the impact of $F_\perp$ from an empirical and global perspective. While InfoNCE aims to push text embeddings away from negative samples, two key factors suggest that the perpendicular component is suboptimal for this goal:

- Mini-Batch Stochasticity: Current InfoNCE implementations rely on Mini-Batch training. Consequently, the direction of the pushing force within a specific mini-batch inevitably deviates from the true gradient direction of the entire dataset. This stochastic deviation acts as noise in cross-modal representation alignment. This is supported by the consensus in contrastive learning research that larger batch sizes—which reduce gradient variance—consistently yield better alignment performance.

- Global Trajectory Drift: Even from a global perspective, the aggregate pushing force generated by the distribution of all negative samples is seldom perfectly collinear with the pulling force from the positive sample. This generates a Global Perpendicular Component 1. Although this component geometrically pushes the embedding away from negative samples, it simultaneously pushes the optimization path sideways, deviating from the ideal straight-line trajectory required to align with the positive sample.

# D  Analysis of SVR's Mechanism

In this section, we analyze the gradient space of the SVR term to show that SVR can effectively suppress the perpendicular component and improve the accuracy of the learned representations in theory.

The support vector is located in the direction pointing from text embedding $t^+$ to positive audio embedding $a^+$ and lies on the surface of the text embedding distribution centered at text embedding $t^+$. Its mathematical expression is $t_{sup} = t^+ + R\frac{a^+-t^+}{||a^+-t^+||} = t^+ + R\hat{u}$, where $\hat{u} = \frac{a^+-t^+}{||a^+-t^+||}$ is the unit vector in the pulling direction and $R$ is an adaptive and learnable radius.

The gradient of SVR term in equation 6 is computed by the chain rule:

$$\nabla_t L_{svr} = \left(\frac{\partial t_{sup}}{\partial t^+}\right)^T \nabla_{t_{sup}} L_{svr}. \tag{13}$$

The Jacobian matrix $\frac{\partial t_{sup}}{\partial t^+}$ is computed as

$$\begin{aligned} \frac{\partial t_{sup}}{\partial t^+} &= \frac{\partial}{\partial t^+}\left(t^+ + R\frac{a^+-t^+}{||a^+-t^+||}\right) \\ &= I + R\frac{\partial}{\partial t^+}\left(\frac{a^+-t^+}{||a^+-t^+||}\right). \end{aligned} \tag{14}$$

After derivation, we get $\frac{\partial \hat{u}}{\partial t^+} = \frac{1}{||a^+-t^+||}(\hat{u}\hat{u} - I)$. The matrix $(\hat{u}\hat{u} - I)$ is a projection operation that projects any vector onto the hyperplane orthogonal to $\hat{u}$. We denote $P_\perp = I - \hat{u}\hat{u}$. Therefore, the Jacobian matrix is denoted as

$$\frac{\partial t_{sup}}{\partial t^+} = I - \frac{R}{||a^+-t^+||}P_\perp. \tag{15}$$

We further analyze the pushing force component from a single negative example $a_j^-$, denoted as $f_{sup,push,j}$, within the gradient generated by $L_{svr}$ term:

$$f_{sup,push,j} = \alpha \left( \frac{\partial t_{sup}}{\partial t^+} \right)^T \left( \frac{P_{sup,j}^-}{\tau} a_j^- \right)$$

$$= \alpha \frac{P_{sup,j}^-}{\tau} \left( a_j^- - \frac{R}{||a^+ - t^+||} P_\perp a_j^- \right)$$

$$= \alpha \frac{P_{sup,j}^-}{\tau} \left( (a_{||,j}^- + a_{\perp,j}^-) - \frac{R}{||a^+ - t^+||} a_{\perp,j}^- \right) \qquad (16)$$

$$= \alpha \frac{P_{sup,j}^-}{\tau} \left( a_{||,j}^- + (1 - \frac{R}{||a^+ - t^+||}) a_{\perp,j}^- \right),$$

where $\alpha$ denotes the weight of the SVR term in the total loss, probability $P_{sup,j}^-$ can be calculated in equation 3 by replacing text embedding $t^+$ to text support vector $t_{sup}$. We decompose the negative audio embedding $a_j^-$ into a parallel component $a_{||,j}^-$ and perpendicular component $a_{\perp,j}^-$ to the unit vector $\hat{u}$, denoted as $a_j^- = a_{||,j}^- + a_{\perp,j}^-$. It should be noted that $P_\perp a_j^- = (I - \hat{u}\hat{u}) a_j^- = a_{\perp,j}^-$.

This result is the core of the proof. It shows that for the pushing force generated by the support vector loss, its parallel component $a_{||,j}^-$ remains unchanged, while its perpendicular component $a_{\perp,j}^-$ is scaled by a factor of $(1 - \frac{R}{||a^+ - t^+||})$.

We examine the parallel component of the total pushing force from negative example $j$, denoted as $F_{||,push,j}$:

$$F_{||,push,j} = f_{||,j} + f_{sup,||,j}$$

$$= \frac{P_j^-}{\tau} a_{||,j}^- + \alpha \frac{P_{sup,j}^-}{\tau} a_{||,j}^- \qquad (17)$$

$$= \left( \frac{P_j^-}{\tau} + \alpha \frac{P_{sup,j}^-}{\tau} \right) a_{||,j}^-.$$

And the perpendicular component $F_{\perp,push,j}$ of the total pushing force from negative example $j$ can be computed as:

$$F_{\perp,push,j} = f_{\perp,j} + f_{sup,\perp,j}$$

$$= \frac{P_j^-}{\tau} a_{\perp,j}^- + \alpha \frac{P_{sup,j}^-}{\tau} \left( 1 - \frac{R}{||a^+ - t^+||} a_{\perp,j}^- \right) \qquad (18)$$

$$= \left[ \frac{P_j^-}{\tau} + \alpha \frac{P_{sup,j}^-}{\tau} \left( 1 - \frac{R}{||a^+ - t^+||} \right) \right] a_{\perp,j}^-.$$

The factor $(1 - \frac{R}{||a^+ - t^+||})$ plays a key modulating role to control the optimization trajectory drift. For most of the training process, the text embedding $t^+$ is at a certain distance from its positive example $a^+$, such that $||a^+ - t^+|| > R$. In this scenario, the factor is a positive number less than 1, which means the support vector regularization term actively weakens this drift caused by the perpendicular component. Consequently, the overall gradient direction points more closely toward the pulling force while adaptively maintaining enough additional information in negative samples, making the optimization path from text embedding $t^+$ to positive audio embedding $a^+$ more direct and stable. This reduces unnecessary drift, thereby accelerating the convergence process and potentially leading to better alignment.

# E  ADDITIONAL EXPERIMENT

## E.1  STATISTICAL SIGNIFICANCE EVALUATION

To ensure the robustness of our results and verify that the performance gains are statistically significant rather than artifacts of random initialization, we performed 5 independent training runs using different random seeds for the InfoNCE, SigLIP, and our proposed variants. We report the mean and standard deviation of the retrieval metrics on the AudioCaps dataset.

As presented in Table 4, our SVR-based methods consistently outperform the baselines with low variance. For instance, compared to the strong InfoNCE baseline ($41.79 \pm 0.14$), our bi-DynamicSVR achieves a T2A R@1 of $44.29 \pm 0.61$. Notably, the lower bound of our method's confidence interval is strictly higher than the upper bound of the baseline, indicating that the improvement is statistically significant. A similar trend is observed when comparing our method against SigLIP, where bi-DynamicSVR yields a substantial improvement of over 6% in T2A R@1. These results confirm that controlling the perpendicular component of the pushing force leads to stable and reproducible performance gains.

Table 4: Performance comparison on AudioCaps with mean and std.

| Method | Text-to-Audio | | Audio-to-Text | |
|---|---|---|---|---|
| | **R@1** | **R@10** | **R@1** | **R@10** |
| SigLIP | $36.68_{\pm 0.12}$ | $85.83_{\pm 0.15}$ | $47.92_{\pm 0.11}$ | $88.14_{\pm 0.19}$ |
| - bi-StaticSVR | $42.67_{\pm 0.28}$ | $87.55_{\pm 0.34}$ | $55.41_{\pm 0.25}$ | $90.42_{\pm 0.31}$ |
| - bi-DynamicSVR | $\mathbf{43.21}_{\pm 0.67}$ | $\mathbf{89.15}_{\pm 0.58}$ | $\mathbf{56.44}_{\pm 0.72}$ | $\mathbf{92.81}_{\pm 0.63}$ |
| InfoNCE | $41.79_{\pm 0.14}$ | $87.75_{\pm 0.16}$ | $56.61_{\pm 0.13}$ | $92.41_{\pm 0.18}$ |
| - bi-StaticSVR | $43.76_{\pm 0.33}$ | $88.92_{\pm 0.29}$ | $57.64_{\pm 0.37}$ | $92.88_{\pm 0.24}$ |
| - bi-DynamicSVR | $\mathbf{44.29}_{\pm 0.61}$ | $\mathbf{89.08}_{\pm 0.53}$ | $\mathbf{59.83}_{\pm 0.79}$ | $\mathbf{93.35}_{\pm 0.68}$ |

## E.2  EXPERIMENT RESULT UNDER LARGE-SCALE SCENARIO AND DISTRIBUTION SHIFTS SCENARIO

To further validate the efficacy of SVR in real-world scenarios, we evaluated it under two settings: data scaling and distribution shift. For the data scaling setting, we utilized the WavCaps dataset, which encompasses a broader range of audio events and comprehensively reflects complex real-world data distributions. Specifically, we held out 5,000 samples as the test set and used the remainder for training. regarding the distribution shift scenario, we evaluated the performance of models trained on the Clotho dataset using the AudioCaps dataset, and vice versa.

Table 5: Performance of InfoNCE and InfoNCE+SVR under WavCaps dataset

| Model | A2T | T2A |
|---|---|---|
| | R@1 | R@1 |
| InfoNCE | 20.60 | 7.76 |
| -bi-StaticSVR | 21.00 | 8.26 |
| -bi-DynamicSVR | **21.30** | **8.30** |

As shown in Table 5, both StaticSVR and DynamicSVR consistently outperform the standard InfoNCE baseline. specifically, DynamicSVR achieves the highest accuracy with 21.30% (A2T) and 8.30% (T2A) in R@1. This confirms that our SVR method remains effective and beneficial when applied to large-scale, diverse real-world data.

Table 6: Performance of InfoNCE and InfoNCE+SVR under distribution shifts scenario

| Model | AudioCaps $\rightarrow$ Clotho | | Clotho $\rightarrow$ AudioCaps | |
|---|---|---|---|---|
| | A2T | T2A | A2T | T2A |
| | R@1 | R@1 | R@1 | R@1 |
| InfoNCE | 18.1643 | 14.4541 | 26.6807 | 20.6513 |
| -bi-StaticSVR | 19.4203 | 14.8213 | 28.3613 | 21.0924 |
| -bi-DynamicSVR | **20.1932** | **14.9279** | **29.6218** | **21.3655** |

Furthermore, we assessed the model's robustness to domain differences through cross-dataset evaluation, as shown in Table 6. In this zero-shot setting—training on AudioCaps and testing on Clotho, and vice versa—our SVR-based methods demonstrate significant performance gains over the baseline. Notably, DynamicSVR improves A2T R@1 by approximately 2.0% and 2.9% in the two transfer scenarios, respectively. These results suggest that by controlling optimization trajectory drift, our SVR methods learn more universal representations that generalize well to unseen domains with distinct data characteristics.

### E.3 DISTRIBUTION OF POSITIVE PAIR SIMILARITY

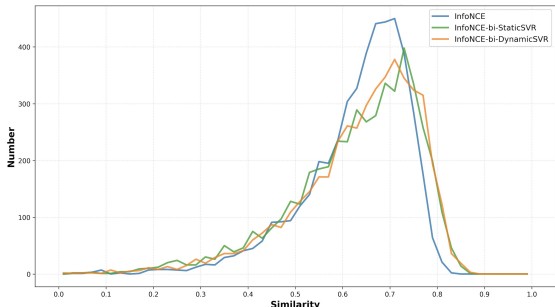

Figure 4: **Distribution of Positive Pair Similarity in AudioCaps**
.

To further demonstrate the robustness of our approach, we analyzed the similarity score distribution of positive pairs on the AudioCaps test set, as shown in Figure 4. The blue curve represents the baseline InfoNCE model, while the green and orange curves correspond to our InfoNCE-bi-StaticSVR and InfoNCE-bi-DynamicSVR methods, respectively.As illustrated, the baseline InfoNCE distribution peaks earlier (around 0.65–0.70) and decays rapidly in the high-similarity region ($> 0.75$). In contrast, both SVR-based methods exhibit a noticeable rightward shift in the distribution. Specifically, our methods maintain a significantly higher density of positive pairs in the high-confidence interval (similarity scores between 0.7 and 0.9) compared to the baseline. This distributional shift indicates that the proposed SVR mechanisms effectively mitigate the semi-hard negative problem by pulling positive pairs closer in the shared embedding space. The increased number of high-similarity pairs provides empirical evidence that the performance improvements are driven by superior and more tightly clustered cross-modal alignment, rather than random variance.

### E.4 COMPARISON SVR WITH SIMPLE GRADIENT REWEIGHTING

To address the question of whether the proposed Support-Vector Regularization (SVR) functions merely as a form of gradient re-scaling, we implemented a baseline method named Simple Reweight under the same setting as SVR for comparison. This analysis serves to demonstrate that the improvements yielded by SVR are not side effects of simple magnitude control, but rather the result of its unique geometric properties.

The Simple Reweight baseline attempts to achieve directional correction by amplifying the weight of the positive sample gradient. We introduced a learnable scalar parameter to scale the probability $P^+$ computed in equation 2. This method emphasizes the parallel component of the gradient relative to the negative samples, theoretically mimicking a basic form of directional adjustment.

We evaluated this baseline on the AudioCaps dataset. As shown in Table 7, the Simple Reweight strategy resulted in performance degradation across both Audio-to-Text (A2T) and Text-to-Audio (T2A) retrieval tasks compared to the standard InfoNCE loss.

The performance degradation in the Simple Reweight baseline highlights a critical distinction in optimization dynamics: the coupling between gradient direction and magnitude.

- Instability of Simple Reweighting: Simply increasing the weight of the positive gradient causes the learnable scalar to simultaneously alter the gradient direction and significantly

Table 7: Performance Comparsion between InfoNCE and Simple Reweight Baseline.

| Model | A2T | T2A |
|---|---|---|
| | R@1 | R@1 |
| InfoNCE | 56.72 | 41.87 |
| -Simple Reweight | 51.26 | 39.15 |

inflate the magnitude of the parallel component. This coupling destabilizes the training process, as the scalar itself becomes difficult to model in an unsupervised manner when the gradient magnitudes fluctuate drastically.

- Stability of SVR: In contrast, SVR employs the scaling factor $(1 - \frac{R}{\|a-t\|})$ to selectively adjust the perpendicular component. Crucially, the unsupervised modeled radius $R$ does not directly inflate the magnitude of the parallel component. This decoupling ensures that the gradients remain stable after regularization, avoiding the magnitude explosion observed in simple re-weighting.

Therefore, while SVR can be theoretically viewed as a sophisticated form of re-weighting, its ability to control trajectory drift without destabilizing optimization is unique and cannot be replicated by simple scalar multiplication.

### E.5 EXPERIMENT RESULT UNDER MULTILINGUAL SCENARIO

Table 8: Recall and precision results under multilingual AudioCaps and Clotho dataset.

| Model | AudioCaps | | | | Clotho | | | |
|---|---|---|---|---|---|---|---|---|
| | T2A | | A2T | | T2A | | A2T | |
| | R@1 | R@10 | R@1 | R@10 | R@1 | R@10 | R@1 | R@10 |
| SigLIP | 34.40 | 82.34 | 44.26 | 86.24 | 13.62 | 50.53 | 17.50 | 53.36 |
| -bi-StaticSVR | 36.87 | 84.23 | 49.54 | 89.08 | 15.17 | 52.32 | 18.70 | 57.00 |
| -bi-DynamicSVR | **38.56** | **85.20** | **51.46** | **90.68** | **15.82** | **54.36** | **21.22** | **58.20** |
| InfoNCE (ML-CLAP) | 37.20 | 84.79 | 50.20 | 90.68 | 17.10 | 55.79 | 21.50 | 58.80 |
| -bi-StaticSVR | 39.60 | 84.98 | 52.36 | 91.05 | 17.28 | 56.06 | 22.37 | 59.42 |
| -bi-DynamicSVR | **39.75** | **85.52** | **53.99** | **91.22** | **17.49** | **56.35** | **22.43** | **60.04** |
| ATRI-CACL | 39.44 | 85.36 | 53.42 | 91.05 | - | - | - | - |
| -bi-StaticSVR | 40.01 | 84.98 | 54.28 | 92.20 | - | - | - | - |
| -bi-DynamicSVR | **43.61** | **89.13** | **60.08** | **94.22** | - | - | - | - |

**Multilingual Retrieval:** Multilingual Retrieval: To further validate the effectiveness of our proposed framework, we conducted evaluations in a multilingual scenario. Following the scheme of ML-CLAP Yan et al. (2024), we dynamically pair each audio sample with a random text caption from one of eight languages during training. The reported metrics are averaged across all languages. We compared our approach against standard losses like SigLIP, InfoNCE, and the recent strong baseline ATRI Yin et al. (2025). For ATRI, we reproduced its Audio-English Co-Anchor Contrastive Learning (CACL) objective.

As shown in Table 8, both bi-StaticSVR and bi-DynamicSVR consistently improve performance across all baselines. Notably, incorporating SVR into ATRI yields significant gains, with bi-DynamicSVR achieving the best overall performance. These results confirm that our SVR mechanism is orthogonal to existing advanced schemes and can be effectively integrated to enhance state-of-the-art models, validating its generalizability even in complex multilingual scenarios. We also evaluate the quality of the translated multilingual test set in Appendix E.9.

### E.6 EXPERIMENT RESULT FOR CONVERGENCE SPEED

To empirically validate that SupCLAP establishes a more effective optimization trajectory, we analyzed the model's convergence speed. Figure 5 illustrates the evolution of the R@1 metric across training epochs for both Audio-to-Text (A2T) and Text-to-Audio (T2A) retrieval tasks on the AudioCaps dataset.

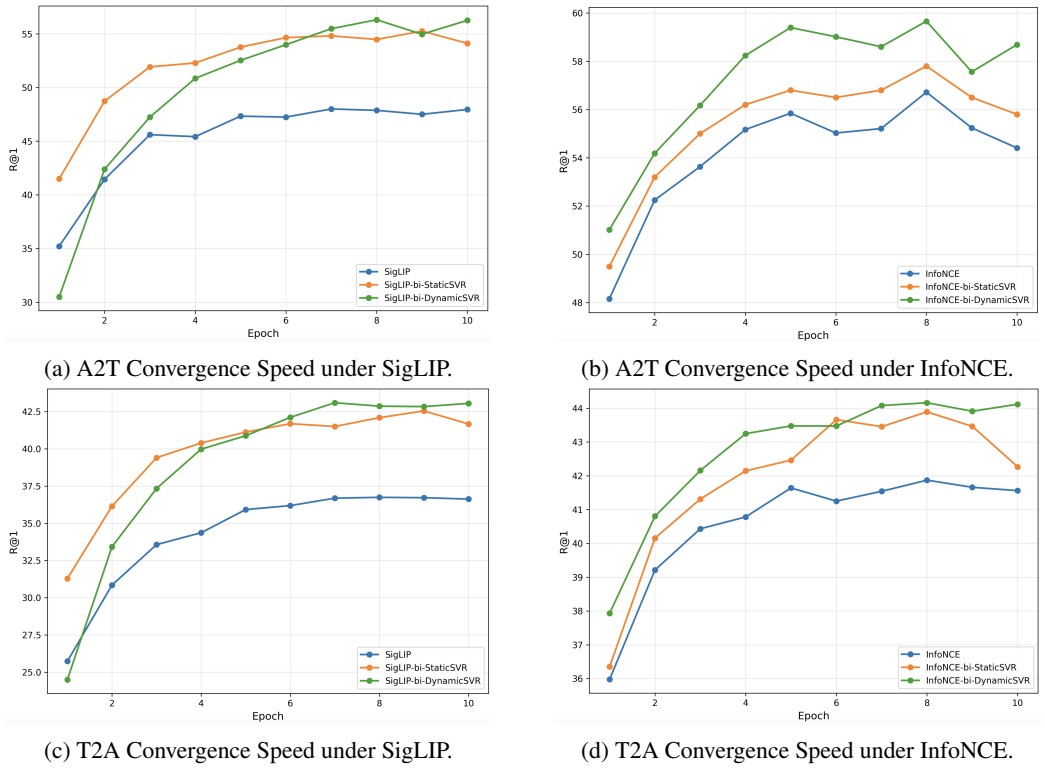

(a) A2T Convergence Speed under SigLIP.

(b) A2T Convergence Speed under InfoNCE.

(c) T2A Convergence Speed under SigLIP.

(d) T2A Convergence Speed under InfoNCE.

Figure 5: Comparison of Convergence Speed between Baseline loss and SVR.

As observed, both bi-StaticSVR and bi-DynamicSVR exhibit significantly faster convergence compared to the standard InfoNCE and SigLIP baselines. The SVR-enhanced models consistently achieve higher retrieval performance at earlier epochs and maintain this advantage throughout the training process. This empirical evidence corroborates our theoretical analysis: by suppressing the perpendicular component of the pushing force, SVR effectively mitigates optimization trajectory drift, resulting in a more direct and efficient path toward optimal alignment.

### E.7 ROBUSTNESS ANALYSIS OF DYNAMICSVR UNDER NOISY SETTINGS

To further explore the stability of the unsupervised dynamic semantic radius predictor under conditions with low-quality or noisy embeddings, we conducted a sensitivity analysis in a specific Noisy Setting.

In this setting, both the audio and text encoders were initialized with random weights instead of pretrained checkpoints. Furthermore, to limit the modeling capacity, we reduced the depth of the text encoder (SONAR-TE) from 24 layers to 12 layers. We compared the performance of the standard InfoNCE baseline against our InfoNCE-bi-DynamicSVR method on the AudioCaps dataset.

The retrieval performance under this noisy setting is reported in Table 9. Despite the lack of initial semantic alignment, the InfoNCE-bi-DynamicSVR method demonstrates significant improvements over the standalone InfoNCE baseline across all metrics.

Table 9: Performance of InfoNCE and InfoNCE-bi-DynamicSVR under noisy setting

| Model | R@1 | | mAP@10 | |
|---|---|---|---|---|
| | A2T | T2A | A2T | T2A |
| InfoNCE | 8.1933 | 7.4160 | 5.2340 | 15.0193 |
| -bi-DynamicSVR | **10.7143** | **9.4328** | **7.0139** | **17.9908** |

The results indicate that DynamicSVR is robust to noisy input embeddings and does not lead to training collapse or instability. To understand the underlying mechanism, we analyzed the evolution of the predicted semantic radius $R$ during training in the noisy setting. According to the result in Figure 6, we observed that the average radius $R$ predicted by the module in the noisy setting is notably larger than that observed when using pre-trained weights.

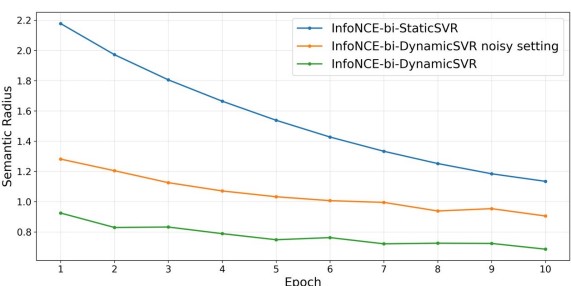

Figure 6: **Variation of Semantic Radius under Noisy Settings and Pre-trained Weights.**

This phenomenon aligns with our theoretical framework regarding optimization trajectory drift. In the noisy setting, there are higher cosine similarities between negative pairs compared to a well-clustered pre-trained space. According to our force decomposition analysis, higher negative similarities result in a significantly larger perpendicular component of the pushing force. The DynamicSVR module correctly identifies this instability and adaptively predicts a larger radius $R$. This increased radius exerts a stronger suppression effect on the perpendicular component (via the scaling factor $1 - R/\|a^+ - t^+\|$), thereby preventing excessive optimization trajectory drift even when the encoder signals are weak. This confirms that the unsupervised predictor functions as an effective adaptive regularizer, scaling its intervention based on the quality of the embedding space.

### E.8 IMPACT OF SEMANTIC DIFFICULTY ON SELF-SUPERVISED SEMANTIC RADIUS MODELING

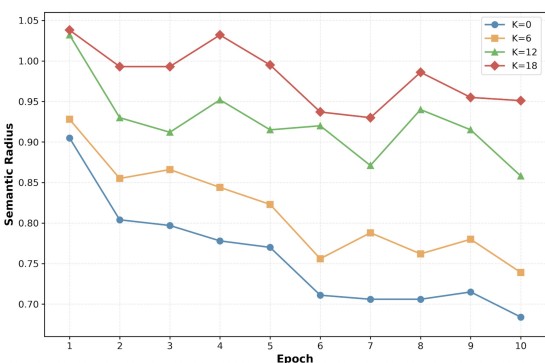

Figure 7: **Variation of Semantic Radius under Different Semantic Difficulties.**

To elucidate the mechanism of the unsupervised dynamic semantic radius predictor and demonstrate its robustness, we conducted an experiment analyzing the variation of the predicted semantic radius $R$ in the presence of hard negative samples. Specifically, we randomly sampled 720 negative samples and identified the Top-K (where $K \in \{6, 12, 18\}$) most similar ones as "hard negatives." These hard negatives were then mixed with general negatives and positive pairs within a mini-batch to monitor the changes in the semantic radius $R$ predicted by the DynamicSVR module across different epochs.

As illustrated in Figure 7, we observed a clear positive correlation: the magnitude of the predicted radius $R$ consistently increases as the density of hard negative samples ($K$) rises. This empirical finding provides a precise mechanistic explanation for the predictor's behavior, countering concerns regarding its fragility:

- Response to Semantic Difficulty: The high similarity characteristic of hard negative samples generates a significantly stronger perpendicular component in the gradient space compared to easy negatives. According to our theoretical analysis, this increases the risk of drift in the optimization trajectory. The radius predictor robustly detects this condition and adaptively increases $R$. A larger $R$ strengthens the regularization effect (via the scaling factor $1 - R/\|a^+ - t^+\|$) to impose tighter control, thereby mitigating the risk of excessive optimization path deviation and ensuring stability.

- Temporal Evolution: As training progresses and the model's discriminative capability improves, the cosine similarity between negative pairs naturally decreases. This attenuates the pushing force and the associated perpendicular component derived from negative samples. Consequently, the predictor adaptively decreases $R$ to avoid over-suppressing a component that is already minimal.

These results confirm that the unsupervised radius predictor successfully learns a dynamic trade-off: it prioritizes stability by assigning a larger $R$ when the risk of drift is high (e.g., early training stages or hard negative presence) and prioritizes information retention by reducing $R$ when the optimization path becomes clearer. This validates that our module is not fragile under noise but rather acts as an adaptive regulator for the perpendicular component.

### E.9 QUALITY ASSESSMENT OF MULTILINGUAL TEST SET

Table 10: Semantic Similarity between Re-translated Text and Original English Text on Test Set

| Language | AudioCaps | Clotho |
|----------|-----------|--------|
| French   | 94.5      | 93.5   |
| Dutch    | 95.0      | 94.2   |
| Spanish  | 94.8      | 94.0   |
| German   | 95.6      | 94.2   |
| Catalan  | 91.1      | 92.2   |
| Japanese | 91.7      | 90.9   |
| Chinese  | 90.7      | 90.3   |

To verify the semantic fidelity of our LLM-translated multilingual Audiocap and Clotho test set and thus substantiate our experimental results, we performed a back-translation analysis. We first translated the test captions back to English using Deepseek V3. We then computed the average cosine similarity between the embeddings of the back-translated and original English texts via the Roberta-large model. The results, detailed in Table 10, reveal an average similarity exceeding 90% for all languages. This high fidelity demonstrates strong semantic preservation and suggests that translation errors in our test set are sufficiently small.

### E.10 ADDITIONAL ABLATION STUDY

Table 11: Ablation Study of SVR Variants on Multilingual Text-Audio Retrieval.

| ID | Model | T2A | | A2T | |
|----|-------|-----|-----|-----|-----|
| | | R@1 | mAP10 | R@1 | mAP10 |
| 0 | InfoNCE | 37.20 | 52.19 | 50.20 | 28.28 |
| 1 | -bi-DynamicSVR | **39.75** | **54.22** | **53.99** | **32.82** |
| 2 | -bi-DynamicSVR wo/ constraints | 39.59 | 54.16 | 53.39 | 32.42 |
| 3 | -uni-DynamicSVR | 39.52 | 54.08 | 53.26 | 32.26 |
| 4 | -uni-DynamicSVR wo/ constraints | 39.37 | 53.81 | 53.07 | 31.78 |
| 5 | -bi-StaticSVR | 39.60 | 53.99 | 52.36 | 31.59 |
| 6 | -uni-StaticSVR | 39.51 | 53.77 | 52.06 | 30.70 |

**Effectiveness of SVR Components:** The multilingual ablation study results in Table 11 are consistent with the monolingual scenario in Table 3; in both cases, the bidirectional DynamicSVR model with constraints (bi-DynamicSVR) achieves the best results. The performance ranking from highest to lowest is bidirectional SVR, unidirectional SVR, and InfoNCE.

Table 12: Ablation study on different batch sizes for T2A and A2T tasks

| Model | Batch Size | T2A | | A2T | |
|---|---|---|---|---|---|
| | | R@1 | mAP10 | R@1 | mAP10 |
| InfoNCE | 24 | 37.20 | 52.19 | 50.20 | 28.28 |
| | 48 | 39.32 | 53.95 | 53.28 | 31.99 |
| | 72 | **40.22** | **54.85** | **54.34** | **32.93** |
| -bi-StaticSVR | 24 | 39.60 | 53.99 | 52.36 | 31.59 |
| | 48 | 40.54 | 55.01 | 54.21 | 32.92 |
| | 72 | **40.89** | **55.40** | **54.87** | **33.47** |
| -bi-DynamicSVR | 24 | 39.75 | 54.22 | 53.99 | 32.82 |
| | 48 | 40.83 | 55.20 | 54.31 | 33.25 |
| | 72 | **40.92** | **55.57** | **55.00** | **33.82** |

**Impact of Batch Size:** Table 12 shows that the performances of both InfoNCE, bi-StaticSVR, and bi-DynamicSVR improve as the batch size increases (24, 48, 72). This is consistent with the principle that larger batches provide more diverse negative samples for contrastive learning, which in turn leads to better performance. Both bi-StaticSVR and bi-DynamicSVR consistently maintain a stable performance advantage over InfoNCE, indicating that our methods can further enhance the performance of InfoNCE with larger batch sizes.

Table 13: Ablation of SVR weight $\alpha$

| $\alpha$ | T2A | | A2T | |
|---|---|---|---|---|
| | R@1 | mAP10 | R@1 | mAP10 |
| 0.1 | 38.69 | 53.38 | 52.05 | 30.48 |
| 0.5 | 39.42 | 53.90 | 52.28 | 31.43 |
| 1.0 | **39.75** | **54.22** | **53.99** | **32.82** |

**Impact of SVR Weight $\alpha$:** The results in Table 13 indicate that model performance consistently improves as $\alpha$ increases from 0.1 to 1.0. This suggests that a greater weight for the SVR module significantly enhances the model's feature representation or matching capabilities. Therefore, the experiment identifies $\alpha = 1.0$ as the optimal hyperparameter setting for the current configuration.

### E.11 Overhead Analysis

Table 14: Evaluation results in GPU memory overheads and time overheads

| Scheme | AudioCaps | | Clotho | |
|---|---|---|---|---|
| | GMO(MB) | TO(s) | GMO(MB) | TO(s) |
| InfoNCE | 22724 | 2479 | 30318 | 1102 |
| -bi-StaticSVR | 22732 | 2438 | 30368 | 1127 |
| -bi-DynamicSVR | 22834 | 2519 | 30398 | 1116 |

In Table 14, we evaluate the time overhead (TO) and GPU memory overhead (GMO) of SupCLAP compared to InfoNCE. TO denotes the average training time over 10 epochs, and GMO denotes the peak GPU memory usage during the training process. The results show that the two SupCLAP variants achieve performance gains with almost no additional computational overhead, possessing both high efficiency and practical viability.

## F Statement on The Use of Large Language Models

We utilized a large language model (LLM) as a general-purpose writing assistant to enhance the clarity and logical structure of the manuscript. This included polishing sentences, improving transitions, and refining the overall flow. All LLM-assisted edits in this paper were meticulously checked and validated by us.

Furthermore, we employed an LLM to construct a multilingual AudioCaps and Clotho dataset. To validate the quality of the translations, we evaluated the semantic similarity between the original En-

glish texts and their translated counterparts in Appendix E.9. This was accomplished by calculating the cosine similarity of their embeddings, which were generated using a RoBERTa-large model.

