# OpenReview forum: "SupCLAP: Controlling Optimization Trajectory Drift in Audio-Text Contrastive Learning with Support Vector Regularization"
_ICLR.cc/2026/Conference — ICLR 2026 Poster_

### Official Review · Reviewer_EVJV · 2025-10-29

**Soundness:** 2
**Presentation:** 3
**Contribution:** 3
**Rating:** 4
**Confidence:** 3

**Summary:**

This paper studies contrastive language–audio pretraining (CLAP) and identifies a phenomenon called optimization trajectory drift, caused by the perpendicular component of the negative-sample pushing force in contrastive learning. The authors propose Support Vector Regularization (SVR), which introduces auxiliary support vectors and a semantic radius to selectively suppress this perpendicular component while preserving useful negative-sample information. They present both static and dynamic strategies for modeling the semantic radius. Experiments across monolingual and multilingual audio–text retrieval, as well as zero-shot classification, show consistent improvements over InfoNCE and SigLIP with negligible overhead.

**Strengths:**

1. The work offers a fresh angle by decomposing contrastive gradients into pulling and perpendicular pushing components, which provides new insights into training instability in CLAP.
2. The SVR formulation is presented with solid mathematical justification, and the derivations clearly show how trajectory drift is effectively suppressed.
3. Experiments across retrieval, classification, and multilingual tasks, along with ablations, demonstrate consistent improvements while keeping the computational cost minimal.

**Weaknesses:**

1. While the paper offers a new analytical angle, the proposed SVR is similar to a regularization tweak of InfoNCE and conceptually close to earlier margin-based or support-vector–style methods.

2. The dynamic semantic radius predictor is unsupervised and fragile under noise. Its meaning is not clearly explained, and why it shrinks during training or how it reflects semantic difficulty is described more heuristically than rigorously.

3. The baseline setup makes comparisons harder to follow. In Tables 1 and 2, many models are cited only by reference instead of name. Some strong recent baselines are also missing, including DS-CLAP (Liu et al., 2024), Cacophony (Zhu et al., 2024), and ATRI (Yin et al., 2025).

4. The experiments are limited to AudioCaps and Clotho, with multilingual results relying on machine translation. Compared with recent work that uses larger datasets such as AudioSet, WavCaps, and LAION-Audio and covers broader tasks like captioning or grounding, the generalization claims are less convincing.


[a] Liu et al., 2024, DSCLAP: Domain-specific contrastive language-audio pre-training

[b] Zhu et al., 2024, Cacophony: An improved contrastive audio-text model

[c] Yin et al., 2025, ATRI: Mitigating multilingual audio-text retrieval inconsistencies by reducing data distribution errors

**Questions:**

1. The dynamic semantic radius relies on an unsupervised predictor. How robust is it under noisy embeddings (e.g., low-resource or weak encoders)? Can the authors show sensitivity or variance analysis to demonstrate stability?

2. The radius shrinks during training, but the explanation is heuristic. Can the authors provide clearer evidence that its values correlate with semantic difficulty (e.g., hard-negative density or retrieval errors)?

---

> ### Author Response · Authors · 2025-11-21
> **Response to Reviewer EVJV**
>
> We sincerely appreciate your comprehensive and rigorous review, particularly regarding the distinction between our method and prior arts, as well as the discussion on robustness across different settings. To address your concerns, we have clarified the fundamental differences between SupCLAP and margin-based methods, incorporated the missing strong baselines (e.g., Cacophony, ATRI), investigated the relationship between semantic difficulty and the semantic radius $R$, and extended our experiments to the WavCaps dataset and noisy settings. These revisions have significantly strengthened our paper. We have incorporated these additional results and analyses into the revised paper, highlighting them in blue; please refer to our detailed responses below for the specific locations.
>
>
>
> **Question 1**: While the paper offers a new analytical angle, the proposed SVR is similar to a regularization tweak of InfoNCE and conceptually close to earlier margin-based or support-vector–style methods.
>
> **Response 1**:
>
> Although our method shares terminological similarities such as support vector with previous related works, there are fundamental differences in motivation, the definition of support vectors, and the underlying mathematical mechanisms. We have carefully compared SupCLAP with representative works in this field, specifically **Max-Margin Contrastive Learning (MMCL)** [1] and **SV-Learner** [2]. The key distinctions ar]e detailed below:
>
> **From the perspective of Support Vector Construction:**
>
> - **MMCL** identifies hard negative samples located on the decision boundary within a batch by solving a quadratic optimization problem, treating these specific samples as support vectors that define the margin.
> - **SV-Learner** mines high-confidence clean samples near the distribution boundary as reliable anchors, primarily used for filtering noisy labels.
> - **SupCLAP**, in contrast, does not select real samples from the dataset as text support vectors ($t_{sup}$). Instead, $t_{sup}$ is a virtually constructed point ($t_{sup} = t^+ + R \cdot \hat{u}$), generated by displacing the original text embedding along the direction of the pulling force, designed to reshape the gradient field rather than serving as a sample selection strategy.
>
> **From the perspective of Core Motivation:**
>
> - **MMCL** aims to address the issue of sub-optimal negative sample selection. Its goal is to maximize the decision margin between positive samples and mined hard negatives to improve sample efficiency.
> - **SV-Learner** addresses the Learning with Noisy Labels problem. It utilizes SVMs to distinguish between clean and noisy samples, preventing erroneous samples from corrupting the contrastive loss.
> - **SupCLAP** focuses on the optimization dynamics itself, specifically addressing the Optimization Trajectory Drift caused by the perpendicular component of the pushing force from negative samples. Our objective is not to select better negatives or filter noise, but to rectify the gradient direction derived from all negative samples to mitigate the issue of optimization path drift.
>
> [1] Shah et al., 2022, Max-Margin Contrastive Learning.
>
> [2] Liang et al., 2024 SV-Learner: Support-Vector Contrastive Learning for Robust Learning With Noisy Labels.

---

> > ### Author Response · Authors · 2025-11-21
> >
> > **Question 2**: The dynamic semantic radius predictor is unsupervised and fragile under noise. Its meaning is not clearly explained, and why it shrinks during training or how it reflects semantic difficulty is described more heuristically than rigorously.
> >
> > **Response 2**:
> >
> > We appreciate the reviewer highlighting the need for a more rigorous explanation of the radius predictor. To elucidate its mechanism and demonstrate its robustness, we conducted an experiment analyzing the variation of the semantic radius $R$ in the presence of hard negative samples. Specifically, we randomly sampled 720 negative samples and selected the Top-$K$ ($K=6, 12, 18$) most similar ones as hard negatives samples. We mixed these hard negatives with general negatives and positive pairs within a mini-batch and monitored the changes in the semantic radius $R$ predicted by the Dynamic predictor across different epochs and $K$ values.
> >
> > As shown in **Figure 7** at **Appendix E.8**, we observed a clear positive correlation: the radius $R$ increases as the density of hard negative samples increases. This empirical finding refines our initial interpretation and provides a precise mechanistic explanation:
> >
> > - **Response to Semantic Difficulty:** The high similarity of hard negative samples generates a stronger perpendicular component in the gradient space, thereby increasing the risk of optimization trajectory drift. The radius predictor robustly detects this condition and increases $R$ to impose stronger regularization. This mitigates the risk of excessive optimization path deviation, thereby ensuring optimization stability.
> > - **Temporal Evolution (Why $R$ shrinks):** As training progresses and the model's discriminative capability improves, the similarity of negative pairs naturally decreases. This attenuates the gradients derived from negative samples. Consequently, in response to the diminished perpendicular component within the gradients, the predictor adaptively decreases $R$ to avoid over-suppressing a component that is already minimal.
> >
> > This confirms that the unsupervised radius predictor is not fragile; rather, it has successfully learned a dynamic trade-off: prioritizing stability (larger $R$) when the risk of drift is high, and prioritizing information retention (smaller $R$) when the optimization path becomes clearer and the perpendicular component is smaller. This aligns with the core proposition of our paper: the perpendicular component is a double-edged sword, and a precise balance must be struck between mitigating optimization trajectory drift and preserving the semantic information from negative samples.

---

> > > ### Author Response · Authors · 2025-11-21
> > >
> > > **Question 3**: The baseline setup makes comparisons harder to follow. In Tables 1 and 2, many models are cited only by reference instead of name. Some strong recent baselines are also missing, including DS-CLAP (Liu et al., 2024), Cacophony (Zhu et al., 2024), and ATRI (Yin et al., 2025).
> > >
> > > **Response 3**:
> > >
> > > We have updated **Tables 1 and 2** by replacing citation indices with explicit model names to enhance clarity. Regarding the suggested baselines, we have incorporated the result of **Cacophony** in **Tables 1** at **Section 4.2** and **ATRI** in **Table 8** at **Appendix E.5**.
> > >
> > > - **Cacophony:** As shown in **Table 1**, SupCLAP outperforms Cacophony on the AudioCaps dataset, though it performs slightly below Cacophony on Clotho. It is crucial to note the substantial disparity in data regimes: Cacophony was trained on approximately 3.9 million pairs (incorporating FreeSound data, a superset of Clotho), whereas our efficient setup utilized only lightweight datasets (AudioCaps or Clotho). Despite this significant data disadvantage, SupCLAP maintains highly competitive performance.
> > >
> > >   **Table 1: Performance of Cacophony and InfoNCE+SVR**
> > >
> > >   |                      | AudioCaps | AudioCaps | Clotho   | Clotho   |
> > >   | -------------------- | --------- | --------- | -------- | -------- |
> > >   |                      | A2T(R@1)  | T2A(R@1)  | A2T(R@1) | T2A(R@1) |
> > >   | Cacophony            | 55.3      | 41.0      | 26.5     | 20.2     |
> > >   | InfoNCE+biStaticSVR  | 57.7      | 43.8      | 24.9     | 19.5     |
> > >   | InfoNCE+biDynamicSVR | 59.6      | 44.1      | 25.3     | 19.7     |
> > >
> > >
> > >
> > > - **DS-CLAP [a]:** We excluded DS-CLAP because it is a domain-specific model trained on private in-car audio data. It is primarily designed for tasks such as speech detection rather than standard open-domain retrieval tasks on AudioCaps or Clotho.
> > > - **ATRI [c]:** To fairly evaluate ATRI within our multilingual setting, we reproduced ATRI's audio-English co-anchor contrastive learning (CACL) method. **Table 2** demonstrates that incorporating SVR into the CACL objective yields further performance improvements. This proves that SVR is orthogonal to existing methods and can be effectively integrated to enhance other strong baselines.
> > >
> > > **Table 2: Performance of ATRI and ATRI+SVR**
> > >
> > > | Methods                   | A2T(R@1)  | A2T(R@10) | T2A(R@1)  | T2A(R@10) |
> > > | ------------------------- | --------- | --------- | --------- | --------- |
> > > | ATRI(CACL)                | 53.42     | 91.99     | 39.44     | 85.36     |
> > > | ATRI(CACL)+biStaticSVR    | 54.28     | 92.20     | 40.01     | 85.92     |
> > > | ATRI((CACL))+biDynamicSVR | **60.08** | **94.22** | **43.61** | **89.13** |
> > >
> > >
> > >
> > > **Question 4**: The experiments are limited to AudioCaps and Clotho, with multilingual results relying on machine translation. Compared with recent work that uses larger datasets such as AudioSet, WavCaps, and LAION-Audio and covers broader tasks like captioning or grounding, the generalization claims are less convincing.
> > >
> > > **Response 4**:
> > >
> > > To validate the generalization capability of our method on larger-scale data, we extended our experimental evaluation to the WavCaps dataset. We held out 5,000 audio-text pairs from WavCaps for testing and utilized the remaining data for training. As presented in **Table 3**, SVR continues to improve the performance of the InfoNCE baseline on this larger and more diverse dataset. This further demonstrates the scalability of SupCLAP and confirms that it maintains its effectiveness in broader, more realistic real-world data regimes beyond standard benchmarks such as AudioCaps and Clotho. We have append the result under WavCaps dataset in **Appendix E.2**.
> > >
> > > **Table 3: Performance of InfoNCE and InfoNCE+SVR under WavCaps dataset**
> > >
> > > |                      | A2T(R1)   | T2A(R1)  |
> > > | -------------------- | --------- | -------- |
> > > | InfoNCE              | 20.60     | 7.76     |
> > > | InfoNCE+biStaticSVR  | 21.00     | 8.26     |
> > > | InfoNCE+biDynamicSVR | **21.30** | **8.30** |

---

> > > > ### Author Response · Authors · 2025-11-21
> > > >
> > > > **Question 5**: The dynamic semantic radius relies on an unsupervised predictor. How robust is it under noisy embeddings (e.g., low-resource or weak encoders)? Can the authors show sensitivity or variance analysis to demonstrate stability?
> > > >
> > > > **Response 5**:
> > > >
> > > > To evaluate robustness under noisy conditions, we conducted experiments where both the audio and text encoders were initialized with random weights. Additionally, we reduced the model complexity by decreasing the number of Transformer layers in the text encoder from 24 to 12. We compared the InfoNCE and InfoNCE+biDynamicSVR methods within this specific setting, and the results are reported in **Table 4**.
> > > >
> > > > **Table 4: Performance of InfoNCE and InfoNCE-bi-DynamicSVR under noisy setting**
> > > >
> > > > | Methods              | A2T(R@1)    | T2A(R@1)   | A2T(mAP10) | T2A(mAP10)  |
> > > > | -------------------- | ----------- | ---------- | ---------- | ----------- |
> > > > | InfoNCE              | 8.1933      | 7.4160     | 5.2340     | 15.0193     |
> > > > | InfoNCE+biDynamicSVR | **10.7143** | **9.4328** | **7.0139** | **17.9908** |
> > > >
> > > > The results demonstrate that InfoNCE+biDynamicSVR significantly outperforms the standalone InfoNCE baseline. This indicates that even with low-quality input embeddings, DynamicSVR remains capable of generating stable semantic radii that facilitate convergence, rather than leading to training collapse or instability.
> > > >
> > > > We further analyzed the evolution of the semantic radius $R$ under this Noisy Setting, as illustrated in **Figure 6** at **Appendix E.7**. The analysis reveals that the radius $R$ predicted by the module in the noisy setting is notably larger than that observed when using pre-trained weights. This finding is consistent with the conclusion presented in **Response 2**: in the noisy setting, the similarity among negative pairs is higher, resulting in a larger perpendicular component. Consequently, the radius predictor adaptively increases $R$ to suppress this component, thereby preventing excessive optimization trajectory drift.
> > > >
> > > >
> > > >
> > > > **Question 6**: The radius shrinks during training, but the explanation is heuristic. Can the authors provide clearer evidence that its values correlate with semantic difficulty (e.g., hard-negative density or retrieval errors)?
> > > >
> > > > **Response 6**: Please see **Response 2**.

---

> ### Author Response · Authors · 2025-11-26
>
> Dear Reviewer EVJV,
>
> We hope this email finds you well. We want to follow up and see if there are any additional questions or concerns about our paper or the rebuttal we provided earlier that we could help clarify.
>
> Your detailed feedback has been Insightful, and we have put significant effort into addressing the points you raised. If there are any remaining aspects where further clarification might strengthen your understanding of our work, we would be willing to provide more information.
>
> Furthermore, if you feel our responses have addressed your concerns effectively, we would greatly appreciate it if you would consider revisiting your initial rating of our submission. Your expert evaluation plays a crucial role in shaping the final outcome, and we sincerely appreciate your time and efforts throughout this review process.
>
> Thank you again for your dedication to improving the quality of submissions. Please feel free to let us know if there is anything else we can assist with.
>
> Best regards,
>
> Authors of Submission 7917

---

### Official Review · Reviewer_pCqQ · 2025-10-29

**Soundness:** 2
**Presentation:** 3
**Contribution:** 2
**Rating:** 4
**Confidence:** 2

**Summary:**

The paper proposes SupCLAP with a Support Vector Regularization (SVR) that shifts the text embedding towards its positive audio.They add an extra contrastive term, claiming to suppress the perpendicular component of the negative “pushing force” and reduce optimization trajectory drift.  And this will help improve performance.

**Strengths:**

This paper gives a novel regularization term on the InfoNCE loss,  their experiments show that this regularization indeed bring benefits. They also provide their intuition in section 2.  And The writing of this paper is good.

**Weaknesses:**

1. In the section 2, the authors discuss the optimization trajectory drift. However, the causal link “lower perpendicular component → better performance” is not theoretically or empirically established. So this section is not very convincing to me. The author may provide theoretical or empirical evidence.
2. For the InfoNCE loss, we originally want the text embedding being away from the negative sample. So why we need to reduce the perpendicular component of the negative “pushing force”?
3. Moreover, the proposed Support-Vector Regularization (SVR) seems closely related to simply re-scaling or re-weighting the contrastive gradients. It would strengthen the paper if the authors compared their method to simpler baselines—e.g., emphasizing the parallel component only or giving positive sample gradient more weights—to demonstrate that the observed improvement is not merely a side-effect of gradient magnitude control.

**Questions:**

See weakness.

---

> ### Author Response · Authors · 2025-11-21
> **Response to Reviewer pCqQ**
>
> We are grateful for your critical questions regarding the theoretical foundation of the optimization trajectory drift and the comparison with simpler baselines. Your feedback prompted us to provide a rigorous mathematical proof and to implement the "Simple Reweight" baseline for comparison. These additions have significantly strengthened the theoretical depth and improved the clarity of our contributions. We have incorporated these additional results and analyses into the revised paper, highlighting them in blue; please refer to our detailed responses below for the specific locations.
>
>
>
> **Question 1**: In the section 2, the authors discuss the optimization trajectory drift. However, the causal link “lower perpendicular component → better performance” is not theoretically or empirically established. So this section is not very convincing to me. The author may provide theoretical or empirical evidence.
>
> **Response 1**:
>
> We elucidate the causal relationship between controlling the perpendicular component and performance improvement by examining the two optimization objectives of contrastive learning: pulling positive pairs closer and pushing negative pairs apart.
>
> Regarding the optimization objective of pulling positive pairs together, we provide a theoretical proof in **Appendix C.1** demonstrating that the perpendicular component reduces the optimization speed of representation alignment in positive pairs. Here, we directly present the formula for computing the convergence speed of positive pairs. A smaller value of the term $||t^+_{next}-a^+||^2-||t^+-a^+||^2$ leads to faster convergence speed:
>
>
> $$
> \begin{aligned}
> ||t_{next}^+-a^+||^2&=||(t^+-a^+)-\eta(F_{||}+F_\perp)||^2  \\\\
> &=||t^+-a^+||^2-2\eta(t^+-a^+)^TF_{||}+\eta^2||F_{||}+F_\perp||^2 \\\\
> &=||t^+-a^+||^2-2\eta(t^+-a^+)^TF_{||}+\eta^2||F_{||}||^2+\eta^2||F_\perp||^2 \\\\
> &=||t^+-a^+||^2-\eta(2t^+-2a^+-\eta F_{||})^TF_{||}+\eta^2||F_\perp||^2, \\\\
> ||t_{next}^+-a^+||^2-||t^+-a^+||^2&=-\eta(2t^+-2a^+-\eta F_{||})^TF_{||}+\eta^2||F_\perp||^2, \\\\
> \end{aligned}
> $$
>
> where $(a^+,t^+)$ denotes the positive pair. $t_{next}^{+}$ denotes the positive text embedding in the next step, computed by $t_{next}^+=t^+-\eta F=t^{+}-\eta(F_{||}+F_\perp)$. Because the learning rate $\eta$ is small enough, the hindrance caused by the parallel component in term $\eta^2 ||F_{||}||^2$ is far smaller than the promotion of convergence speed by the parallel component in term $2\eta(t^+-a^+)^TF_{||}$. Since the term $\eta^2||F_\perp||^2$ is greater than 0, a larger perpendicular component imposes a greater hindrance to convergence speed of positive pairs. This mathematical conclusion is consistent with the intuitive illustration provided in **Figure 1** in **Section 2.1.1**.
>
> Regarding the objective of pushing negative pairs apart, we provide evidence primarily from an empirical perspective. The design of InfoNCE indeed aims to push text embeddings away from negative samples. However, since current InfoNCE implementations rely on Mini-Batch training, the direction of the pushing force within a mini-batch inevitably deviates from that of the entire dataset. This stochastic deviation introduces error into cross-modal representation alignment. The most compelling evidence for this is the consensus in research using InfoNCE loss that larger batch sizes consistently yield better alignment performance. Even from a global perspective, the aggregate pushing force generated by all negative samples is almost never perfectly collinear with the pulling force from the positive sample. This generates a global perpendicular component. Although this component pushes the embedding away from negative samples, it simultaneously pushes the optimization path away from the ideal straight-line trajectory for aligning with the positive sample. SVR does not prevent the model from pushing away negative samples; rather, it acts as a corrector: by suppressing the perpendicular component, it ensures that the force direction aligns more closely with the optimization goal, thereby achieving a more direct and stable optimization trajectory.
>
> We have add the proof and analysis in **Appendix C**.

---

> > ### Author Response · Authors · 2025-11-21
> >
> > **Question 2**: For the InfoNCE loss, we originally want the text embedding being away from the negative sample. So why we need to reduce the perpendicular component of the negative “pushing force”?
> >
> > **Response 2**:
> >
> > Please see **Response 1**
> >
> >
> >
> > **Question 3**: Moreover, the proposed Support-Vector Regularization (SVR) seems closely related to simply re-scaling or re-weighting the contrastive gradients. It would strengthen the paper if the authors compared their method to simpler baselines—e.g., emphasizing the parallel component only or giving positive sample gradient more weights—to demonstrate that the observed improvement is not merely a side-effect of gradient magnitude control.
> >
> > **Response 3**:
> >
> > Our theoretical analysis in **Appendix D** indicates that the core mechanism of SVR can indeed be viewed as a form of gradient re-weighting. Unlike simple static weighting, the weights for gradient scaling in SVR are learnable, and our proposed DynamicSVR is adaptive, predicting a specific radius $R$ for data in different batches. Although SVR can theoretically be viewed as a decoupled and adaptive form of gradient re-weighting, its optimization dynamics are fundamentally distinct from simple scaling methods.
> >
> > To substantiate this, we implemented the suggested baseline named Simple Reweight under AudioCaps dataset, using the same experimental settings as SVR. This baseline amplifies the weight of the positive sample gradient by introducing a learnable scalar to scale the probability $P^+$ (derived from the Softmax of positive similarity), thereby attempting to achieve directional correction similar to SVR. As presented in **Table 1**, this baseline actually resulted in performance degradation compared to InfoNCE.
> >
> > **Table1: Performance of InfoNCE and InfoNCE+SImple Reweight**
> >
> > | Simple Reweight Setting | A2T(R@1) | T2A(R@1) |
> > | ----------------------- | -------- | -------- |
> > | InfoNCE                 | 56.72    | 41.87    |
> > | InfoNCE+Simple Reweight | 51.26    | 39.15    |
> >
> > The reason lies in the coupling between direction and magnitude. Simply increasing the positive gradient causes the learnable scalar to simultaneously alter the gradient direction and significantly inflate the magnitude of the parallel component. This destabilizes both the unsupervised modeling of the scalar and the gradients during contrastive learning, leading to training instability. In contrast, SVR utilizes the scaling factor $(1 - \frac{R}{\|a-t\|})$ to selectively adjust the perpendicular component, where the unsupervised modeled radius $R$ does not directly affect the magnitude of the parallel component. This ensures that the gradients remain stable after unsupervised modeling and SVR regularization, avoiding the gradient magnitude explosion issue associated with simple re-weighting, and thereby securing a stable and effective alignment.
> >
> > We have add the result and analysis in **Appendix E.4**.

---

> ### Author Response · Authors · 2025-11-26
>
> Dear Reviewer pCqQ,
>
> We would like to follow up to see if you have any remaining questions or concerns regarding our paper or our rebuttal response.
>
> We found your detailed feedback extremely insightful and have dedicated significant effort to addressing each of your points. Accordingly, we have updated our manuscript on OpenReview. If there are any lingering issues, or if further explanation would assist you in better understanding our work, we would be more than happy to provide additional information.
>
> Furthermore, if you feel that our responses have effectively addressed your concerns, we kindly ask that you consider reconsidering your rating of our submission. Your professional assessment is crucial to the final outcome, and we sincerely appreciate the time and energy you have invested throughout this review process.
>
> Thank you again for your dedication to improving the quality of submissions. Please feel free to let us know if there is anything else we can assist with.
>
> Best regards,
>
> Authors of Submission 7917

---

### Official Review · Reviewer_nh4f · 2025-10-29

**Soundness:** 3
**Presentation:** 3
**Contribution:** 3
**Rating:** 6
**Confidence:** 3

**Summary:**

This paper presents SupCLAP, a method for regularizing the contrastive learning optimization process of standard objectives (InfoNCE, SigLIP) to improve performance on downstream retrieval and 0-shot classification tasks.

**Strengths:**

- Motivation is reasonably strong, and mathematical rigor to the proposed method is appreciated.
- Despite the overall mathematical maturity of SupCLAP, the proposed method is still intuitive and straightforward, aiding the clarity of the paper.
- The experimental results, while generally modest, show clear gains to the using the proposed method relative to the baselines.

**Weaknesses:**

In general, there is not much massively wrong with the present manuscript, my two biggest concerns are:
- It would be useful to report confidence intervals for the retrieval / classification experiments, as the results are generally modest, and it is hard to tell whether this gain in performance is truly statistically significant.
- Overall, there is not much clear evidence as to *why* SupCLAP seems to work better empirically. While there is ample connection to *in theory* what might be going on during optimization, this is never attempted to be empirically confirmed. It would be very useful to analyze parts of the optimization process for SupCLAP (vs InfoNCE or SigLIP) such as gradient norm, gradient variance, or simply convergence speed to see whether the surmised theoretical reasons for its performance actually translate in practice.

**Questions:**

See weaknesses.

---

> ### Author Response · Authors · 2025-11-21
> **Response to Reviewer nh4f**
>
> We appreciate your insightful suggestions on strengthening the statistical rigor and empirical validation of our work. In response to your comments, we have conducted multiple independent runs to report confidence intervals, ensuring the significance of our results. Furthermore, we have added a detailed convergence analysis and visualized the optimization process to empirically substantiate the theoretical advantages of SupCLAP. We have incorporated these additional results and analyses into the revised paper, highlighting them in blue; please refer to our detailed responses below for the specific locations.
>
>
>
> **Question 1**: It would be useful to report confidence intervals for the retrieval / classification experiments, as the results are generally modest, and it is hard to tell whether this gain in performance is truly statistically significant.
>
> **Response 1**:
>
> We conducted 5 independent training runs using different random seeds for InfoNCE, SigLIP, and our two SVR variants. We calculated the mean and variance of the metrics corresponding to the best R@1 performance. As shown in **Table 1**, the performance improvement achieved by our SVR method over InfoNCE and SigLIP is statistically significant. A detailed analysis of these results is provided in **Appendix E.1**. We also visualized the similarity distribution of positive pairs on the test set of AudioCaps in **Figure 4** at **Appendix E.3**. Compared to the baseline InfoNCE loss, both of our SVR methods exhibit a noticeably higher density in high-similarity regions, which provides strong empirical evidence that SVR achieves superior cross-modal alignment.
>
> **Table 1: Performance comparison on AudioCaps with mean and std.**
>
> | Method                      |           T2A R@1           |          T2A R@10           |           A2T R@1           |          A2T R@10           |
> | :-------------------------- | :-------------------------: | :-------------------------: | :-------------------------: | :-------------------------: |
> | **SigLIP**                  |     $36.68_{\pm 0.12}$      |     $85.83_{\pm 0.15}$      |     $47.92_{\pm 0.11}$      |     $88.14_{\pm 0.19}$      |
> | &nbsp;&nbsp;- bi-StaticSVR  |     $42.67_{\pm 0.28}$      |     $87.55_{\pm 0.34}$      |     $55.41_{\pm 0.25}$      |     $90.42_{\pm 0.31}$      |
> | &nbsp;&nbsp;- bi-DynamicSVR | $\mathbf{43.21}_{\pm 0.67}$ | $\mathbf{89.15}_{\pm 0.58}$ | $\mathbf{56.44}_{\pm 0.72}$ | $\mathbf{92.81}_{\pm 0.63}$ |
> | **InfoNCE**                 |     $41.79_{\pm 0.14}$      |     $87.75_{\pm 0.16}$      |     $56.61_{\pm 0.13}$      |     $92.41_{\pm 0.18}$      |
> | &nbsp;&nbsp;- bi-StaticSVR  |     $43.76_{\pm 0.33}$      |     $88.92_{\pm 0.29}$      |     $57.64_{\pm 0.37}$      |     $92.88_{\pm 0.24}$      |
> | &nbsp;&nbsp;- bi-DynamicSVR | $\mathbf{44.29}_{\pm 0.61}$ | $\mathbf{89.08}_{\pm 0.53}$ | $\mathbf{59.83}_{\pm 0.79}$ | $\mathbf{93.35}_{\pm 0.68}$ |
>
>
>
> **Question 2**: Overall, there is not much clear evidence as to *why* SupCLAP seems to work better empirically. While there is ample connection to *in theory* what might be going on during optimization, this is never attempted to be empirically confirmed. It would be very useful to analyze parts of the optimization process for SupCLAP (vs InfoNCE or SigLIP) such as gradient norm, gradient variance, or simply convergence speed to see whether the surmised theoretical reasons for its performance actually translate in practice.
>
> **Response 2**:
>
> We analyzed the model's convergence speed by tracking the Recall@1 metric across different training epochs for InfoNCE, SigLIP, and our SVR variants. We have presented these convergence results in **Figure  5** in **Appendix E.6** . The results demonstrate that the SVR-enhanced models achieved significantly higher Recall@1 scores at the same training epochs. This empirically confirms that SupCLAP accelerates convergence and establishes a more effective optimization trajectory, thereby corroborating our theoretical analysis.

---

### Official Review · Reviewer_xE2L · 2025-11-01

**Soundness:** 3
**Presentation:** 3
**Contribution:** 3
**Rating:** 6
**Confidence:** 2

**Summary:**

The paper addresses the issue of optimization trajectory drift during contrastive learning due to the perpendicular component of pushing force from negative samples. It highlights how this aspect can lead to instability in training and proposes a novel solution called Support Vector Regularization (SVR).

SVR introduces an auxiliary support vector that allows for better control of negative sample influence, leveraging their informative content while mitigating trajectory drift. The effectiveness of SVR is critically tied to the selection of the semantic radius, and the paper explores two unsupervised modeling strategies: static direct parameterization and an adaptive radius predictor module with constraints.

Extensive experimental results demonstrate that SupCLAP significantly outperforms various baseline methods across tasks such as classification and both monolingual and multilingual retrieval, validating its potential in stabilizing optimization paths and improving model performance.

**Strengths:**

Addresses Optimization Trajectory Drift: The proposed Support Vector Regularization (SVR) method effectively controls optimization trajectory drift caused by the influence of negative samples, enhancing training stability.

Performance Improvement: SupCLAP significantly outperforms various existing baselines across tasks such as classification and monolingual/multilingual retrieval, demonstrating its superior performance in multimodal learning.

Exploration of Unsupervised Modeling Strategies: The study explores both static parameterization and adaptive radius prediction as unsupervised modeling strategies, providing new methodological insights for future research.

**Weaknesses:**

Limited Generalizability Beyond Benchmarks: While the performance in the tested tasks is impressive, the study may have limitations in generalizability. The results may not translate well to all real-world applications, particularly those outside the tested domains or with different data characteristics.

**Questions:**

Given that the SVR method introduces additional parameters that require careful tuning, how does this impact the overall efficiency of the method during training and implementation? Specifically, what are the trade-offs between the improved performance and the potential increases in optimization time?

---

> ### Author Response · Authors · 2025-11-21
> **Response to Reviewer xE2L**
>
> We sincerely appreciate your positive assessment and constructive feedback. Following your valuable suggestions, we have extended our experiments to the larger-scale WavCaps dataset and conducted cross-dataset evaluations to verify the model's robustness under distribution shifts. Furthermore, we have provided a detailed efficiency analysis in the Appendix to address your concerns regarding the trade-offs in computational overhead. We have incorporated these additional results and analyses into the revised paper, highlighting them in blue; please refer to our detailed responses below for the specific locations.
>
>
>
> **Question 1**: Limited Generalizability Beyond Benchmarks: While the performance in the tested tasks is impressive, the study may have limitations in generalizability. The results may not translate well to all real-world applications, particularly those outside the tested domains or with different data characteristics.
>
> **Response 1**:
>
> We have conducted additional experiments on the WavCaps dataset. With a scale approximately eight times that of AudioCaps, WavCaps encompasses a broader range of audio events, offering a more comprehensive reflection of complex real-world data distributions. As shown in **Table 1**, both our StaticSVR and DynamicSVR methods consistently outperform the standard InfoNCE baseline on WavCaps. This demonstrates that our proposed SVR strategies remain effective even when scaled up to larger and more complex data regimes.
>
> **Table1: Performance of InfoNCE and InfoNCE+SVR under WavCaps dataset**
>
> | Methods              | A2T(R@1)  | T2A(R@1) |
> | -------------------- | --------- | -------- |
> | InfoNCE              | 20.60     | 7.76     |
> | InfoNCE+biStaticSVR  | 21.00     | 8.26     |
> | InfoNCE+biDynamicSVR | **21.30** | **8.30** |
>
> Furthermore, to explicitly evaluate the model's robustness against distribution shifts (i.e., out-of-distribution scenarios), we performed cross-dataset evaluations: models trained on AudioCaps were evaluated on the Clotho test set, and vice versa (models trained on Clotho evaluated on AudioCaps). As illustrated in **Table 2**, our SVR methods achieved significant performance gains over the baseline in these zero-shot transfer settings. These results provide strong evidence that by controlling optimization trajectory drift, SupCLAP learns more generalizable representations, thereby enabling effective transfer to scenarios with distinct data characteristics.
>
> **Table 2: Performance of InfoNCE and InfoNCE+SVR under distribution shifts scenario**
>
> |                      | A2T(R@1, AudioCaps train, Clotho test) | A2T(R@1, Clotho train, AudioCaps test) | T2A(R@1, AudioCaps train, Clotho test) | T2A(R@1, Clotho train, AudioCaps test) |
> | -------------------- | -------------------------------------- | -------------------------------------- | -------------------------------------- | -------------------------------------- |
> | InfoNCE              | 18.1643                                | 26.6807                                | 14.4541                                | 20.6513                                |
> | InfoNCE+biStaticSVR  | 19.4203                                | 28.3613                                | 14.8213                                | 21.0924                                |
> | InfoNCE+biDynamicSVR | **20.1932**                            | **29.6218**                            | **14.9279**                            | **21.3655**                            |
>
> We have included the experimental results and analysis for WavCaps and Distribution Shifts Scenario in **Appendix E.2**.
>
>
>
> **Question 2**: Given that the SVR method introduces additional parameters that require careful tuning, how does this impact the overall efficiency of the method during training and implementation? Specifically, what are the trade-offs between the improved performance and the potential increases in optimization time?
>
> **Response 2**:
>
> Experimental results in **Appendix E.11** demonstrate that the additional time overhead and GPU memory overhead introduced by our SVR methods during training are negligible compared to the standard InfoNCE baseline. During inference, we follow conventional approaches by directly computing embeddings through the dual-tower architecture without calculating the SVR term, resulting in no impact on inference overhead. Consequently, there is no significant trade-off between the improved performance and optimization time.

---

### Meta-Review · Area_Chair_JNpy · 2026-01-06

**Summary:**

1.  Reviewer xE2L

a.1 Limited Generalizability Beyond Benchmarks

a.2  The SVR method introduces additional parameters that require careful tuning

2. Reviewer nh4f

b.1 report confidence intervals

b.2 there is not much clear evidence as to why SupCLAP seems to work better empirically

3.  Reviewer pCqQ

c.1 the causal link “lower perpendicular component → better performance” is not theoretically or empirically established. So this section is not very convincing

c.2 why we need to reduce the perpendicular component of the negative “pushing force”

c.3 Need to demonstrate that the observed improvement is not merely a side-effect of gradient magnitude control

4. Reviewer EVJV

d.1 The proposed SVR is similar to a regularization tweak of InfoNCE and conceptually close to earlier margin-based or support-vector–style methods.

d.2 The dynamic semantic radius predictor is unsupervised and fragile under noise.  Why it shrinks during training or how it reflects semantic difficulty is described more heuristically than rigorously.

d.3 The baseline setup makes comparisons harder to follow. Some strong recent baselines are also missing.

d.4 Compared with recent work that uses larger datasets such as AudioSet, WavCaps, and LAION-Audio and covers broader tasks like captioning or grounding, the generalization claims are less convincing.

**Reviewer Concerns:**

1.  Reviewer xE2L

All concerns have been addressed

2. Reviewer nh4f

All concerns have been addressed

3.  Reviewer pCqQ

All concerns have been addressed

4. Reviewer EVJV

All concerns have been addressed

**Reviewer Scores:**

1. Reviewer xE2L will not change the score

2. Reviewer nh4f will not change the score

3. Reviewer pCqQ will increase the score to 6

4. Reviewer pCqQ will increase the score to 6 or not change the score

---

### Decision · Program_Chairs · 2026-01-26

Accept (Poster)